# SpikeVideoFormer: An Efficient Spike-Driven Video Transformer with Hamming Attention and $\mathcal{O}(T)$ Complexity

**Shihao Zou** [1]  **Qingfeng Li** [2]  **Wei Ji** [3]  **Jingjing Li** [4]  **Yongkui Yang** [1]  **Guoqi Li** [2]  **Chao Dong** [1 5]

## Abstract

Spiking Neural Networks (SNNs) have shown competitive performance to Artificial Neural Networks (ANNs) in various vision tasks, while offering superior energy efficiency. However, existing SNN-based Transformers primarily focus on single-image tasks, emphasizing spatial features while not effectively leveraging SNNs' efficiency in video-based vision tasks. In this paper, we introduce SpikeVideoFormer, an efficient spike-driven video Transformer, featuring linear temporal complexity $\mathcal{O}(T)$. Specifically, we design a spike-driven Hamming attention (SDHA) which provides a theoretically guided adaptation from traditional real-valued attention to spike-driven attention. Building on SDHA, we further analyze various spike-driven space-time attention designs and identify an optimal scheme that delivers appealing performance for video tasks, while maintaining only linear temporal complexity. The generalization ability and efficiency of our model are demonstrated across diverse downstream video tasks, including classification, human pose tracking, and semantic segmentation. Empirical results show our method achieves state-of-the-art (SOTA) performance compared to existing SNN approaches, with over 15% improvement on the latter two tasks. Additionally, it matches the performance of recent ANN-based methods while offering significant efficiency gains, achieving $\times 16$, $\times 10$ and $\times 5$ improvements on the three tasks. [1]

## 1. Introduction

SNNs have emerged as an energy-efficient alternative to traditional ANNs, showing significant potential in machine learning (Li et al., 2024). Unlike ANNs, which mainly rely on floating-point multiplications for propagation, SNNs replicate the brain's neural dynamics and 0/1 spike-based communication. This spike-driven mechanism enables SNNs to propagate with simple additions while bypassing non-spiking connections. Consequently, SNNs achieve greater efficiency, offering advantages over ANNs.

Recently, SNNs have demonstrated performance comparable to ANNs across many vision tasks, including image classification (Yao et al., 2024b), object detection (Yao et al., 2024a), semantic segmentation (Wang et al., 2024; Su et al., 2024), and image-text alignment (Li et al., 2023b). While these advancements are significant, current SNN architectures predominantly focus on spatial feature modeling using single-image inputs. This narrow focus overlooks a key attribute of SNNs: their brain-inspired neural dynamics, which inherently supports neuron-level temporal encoding (Li et al., 2024). Such temporal capabilities make SNNs particularly well-suited for video-based vision tasks—a promising yet underexplored area.

Although SNNs support neuron-level temporal encoding, the unidirectional spiking process over time still limits its capability of encoding space-time features in comparison with ANNs. To this end, spike-driven Transformer models (Zhou et al., 2022; Yao et al., 2024a) can be adapted for video-based tasks, enabling space-time feature encoding. However, two key practical issues remain to be addressed: (1) Current spike-driven attention mechanisms directly adopt the dot-product operation from ANN-based attention, which fails to effectively capture the similarity between spike features, as explained in Fig. 2. (2) These spike-driven Transformer models primarily focus on spatial attention, and it remains unclear which spike-driven space-time attention mechanism can properly balance performance and efficiency for SNNs. Although several ANN-based space-time attention designs have been proposed (Arnab et al., 2021), with a particular emphasis on reducing quadratic temporal complexity, these advantages may not be directly applicable to SNNs.

---

[1]Shenzhen Institutes of Advanced Technology, Chinese Academy of Sciences <sh.zou@siat.ac.cn, yk.yang@siat.ac.cn> [2]Institute of Automation, Chinese Academy of Sciences <liqingfeng2023@ia.ac.cn, guoqi.li@ia.ac.cn> [3]Yale University <wei.ji@yale.edu> [4]University of Alberta [5]Shenzhen University of Advanced Technology. Correspondence to: Jingjing Li <jingjin1@ualberta.ca>, Chao Dong <chao.dong@siat.ac.cn>.

[1]https://github.com/JimmyZou/SpikeVideoFormer.

Therefore, in this paper, we introduce SpikeVideoFormer, an efficient spike-driven video Transformer with linear temporal complexity $\mathcal{O}(T)$. At its core, we propose SDHA, which leverages normalized Hamming similarity as the attention score function for spike features. As demonstrated in Proposition 3.1, this function provides a theoretically grounded adaptation from traditional real-valued attention to binary spike-driven attention. While directly applying Hamming similarity poses challenges such as floating-point operations, non-differentiability, and non-linear complexity, we further optimize its design to maintain efficient spike-driven processing. Building on SDHA, we analyze various spike-driven space-time attention designs and identify joint attention as an optimal scheme that delivers appealing performance while maintaining only linear temporal complexity for video tasks. We conduct experiments on three downstream tasks: 1) video classification, a high-level classification task; 2) human pose tracking, a fine-grained regression task; and 3) video semantic segmentation, a dense classification task. Our model achieves SOTA performance among existing SNNs, with over 15% improvements on the latter two tasks. Additionally, our method matches the performance of recent ANN-based methods while increasing efficiency by $\times16$, $\times10$ and $\times5$, respectively.

Our contributions are summarized as follows:

- We introduce SpikeVideoFormer. To our knowledge, we are the first attempt to explore efficient spike-driven video Transformer for video-based vision tasks, featuring linear temporal complexity $\mathcal{O}(T)$.

- We propose spike-driven Hamming attention, providing a theoretically guided adaptation from traditional real-valued attention to spike-driven attention.

- We explore various spike-driven space-time attention variants and provide an optimal design leveraging joint space-time attention, achieving superior performance while with only linear temporal complexity.

- The generalization ability and efficiency of our model are demonstrated across three downstream tasks: video classification, human pose tracking, and video semantic segmentation.

## 2. Related Works

**Convolution-based SNNs** aim to extend the depth and complexity of spiking neural networks by adopting designs of ResNet (He et al., 2016). These models are typically trained using back-propagation through time (BPTT) with surrogate derivatives to approximate the gradient of the spiking function (Zhou et al., 2024). To overcome issues like identity mapping and gradient vanishing, methods such as SEW-SNNs (Fang et al., 2021) and MS-SNNs (Hu et al., 2024) incorporate spike- or membrane-wise shortcut residual connections, achieving performance comparable to ANNs on various simple classification tasks. However, convolution-based SNNs still struggle to deliver competitive performance on many large-scale vision tasks.

**Transformer-based SNNs** represent a significant milestone in advancing spiking neural networks. An earlier study (Yao et al., 2023) introduces multi-dimensional attention for the Spiking Transformer, demonstrating promising performance in large-scale image recognition task. However, this approach relies on real-valued membrane potentials for attention computation, which compromises the efficiency of SNNs. Subsequent works (Zhou et al., 2022; Yao et al., 2024b;a) propose various spike-driven self-attention mechanisms that achieve linear computational complexity w.r.t. the input token length. Despite these advancements, current designs employ Hadamard- or dot-product operations for attention score computation, which theoretically fails to replicate the effectiveness of dot-product score in vanilla Transformer (Vaswani et al., 2017).

**SNNs in Vision Tasks** are crucial for demonstrating their capability in achieving high-efficiency AI. Existing studies have achieved comparable performance with ANNs on various challenging vision tasks, including large-scale image recognition (Fang et al., 2021; Yao et al., 2023; 2024a), semantic segmentation (Su et al., 2024; Wang et al., 2024), and object detection (Su et al., 2023; Luo et al., 2024). Additionally, spiking VAEs (Kamata et al., 2022) and spiking diffusion models (Cao et al., 2024) are proposed for image generation task. SpikeCLIP (Li et al., 2023b) also demonstrates successful alignment between images and text using SNNs. However, existing works only demonstrate the capability of SNNs in static vision tasks, overlooking their potential in more challenging temporal vision tasks.

**Video Transformers** have been widely applied to video vision tasks. A straightforward approach involves concatenating space-time tokens together as input (Wang et al., 2021; Yang et al., 2024). To enhance efficiency, studies have explored alternative attention mechanisms, such as spatiotemporal deformable attention (Wang & Torresani, 2022) and shifted window attention (Liu et al., 2022). Another line of research decouples spatial and temporal attention, either to improve computational efficiency (Arnab et al., 2021) or to take advantage of pre-trained image transformers (Blattmann et al., 2023). Building on these approaches, our work aims to develop an effective spike-driven video transformer, leveraging the unique efficiency of SNNs for video vision tasks.

**Visual Cognitive Neuroscience** studies how factors like attention, motivation, emotion, and expectation shape visual perception and cognition (Oliva & Torralba, 2007). While the first three enhance relevant stimuli processing, expectation suppresses predictable inputs (Vuilleumier, 2005; Summerfield & Egner, 2009). Predictive processing sug-

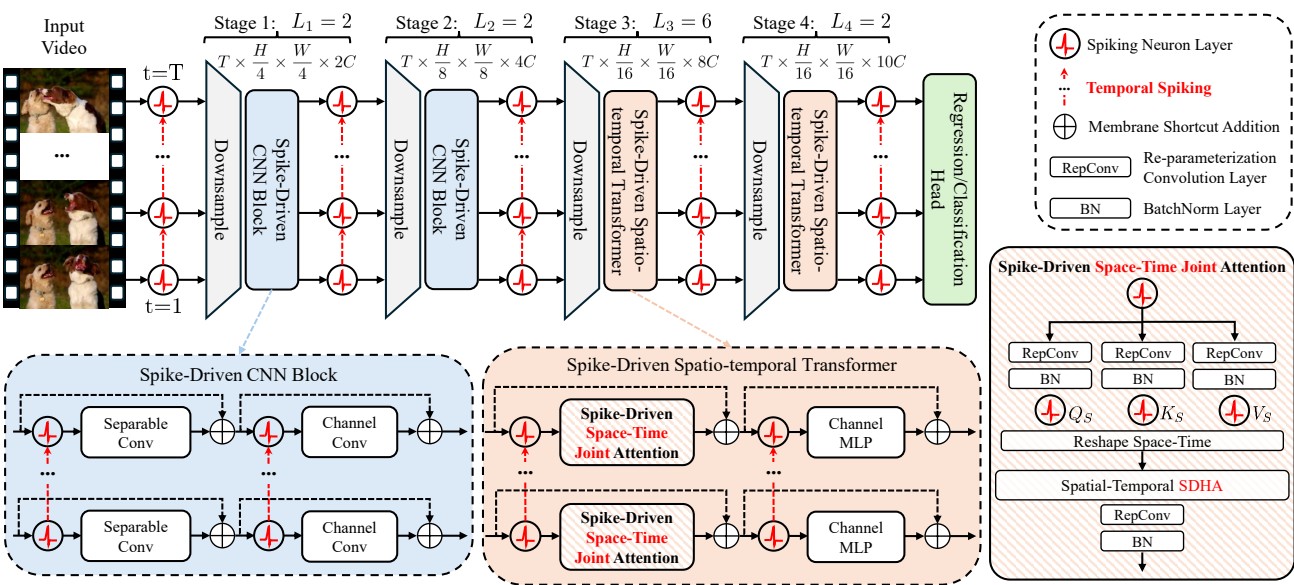

*Figure 1.* **Architecture of our proposed SpikeVideoFormer.** An input video with a shape of $T \times H \times W \times 3$ undergoes temporal spiking over time, after which it passes through two spike-driven CNN blocks, along with downsample modules. This is followed by two spike-driven spatiotemporal transformers, also accompanied by downsample modules. Finally, the extracted video features are forwarded to the regression or classification head for downstream tasks.

gests perception is inference-driven, refining sensory input through internal models shaped by context and experience (Storm et al., 2024). Beyond vision, the visual cortex processes object names and activates in blind individuals, indicating broader cognitive roles in memory, imagery, and language (Bi et al., 2016; Pearson, 2019; Xu, 2017). Though its full scope remains debated, these insights inspire the brain-inspired spiking neural network (SNN) approach for complex visual tasks, enabling high-speed, low-energy neural processing.

## 3. Method

We begin with the preliminaries of the basic spiking neuron model in Sec. 3.1, followed by the overall architecture of SpikeVideoFormer in Sec. 3.2. SpikeVideoFormer consists of two main blocks: (1) a spike-driven CNN block that extracts downsampled spatial features from input videos, and (2) a spike-driven spatiotemporal Transformer that encodes global space-time features for downstream video tasks. Its core functionalities are achieved by the SDHA in Sec. 3.3, which effectively captures the similarity between spike features, and the space-time attention designs in Sec. 3.4, which ensure linear temporal complexity for video processing.

### 3.1. Spiking Neuron Model

The Leaky Integrate and Fire (LIF) model (Maass, 1997) is widely used in SNNs. The LIF neuron maintains a membrane potential $U_{[t]}$ that decays over time according to a leaky constant $\beta$. The potential is updated only when it re-

ceives input spikes $X_{[t]}$ from connected neurons over $T$ time steps. When the membrane potential exceeds a predefined threshold $u_{\text{th}}$, the neuron emits a spike $S_{[t]}$ and undergoes a soft reset reducing its potential by $u_{\text{th}}$. The process is described as follows:

$$H_{[t]} = \beta U_{[t-1]} + X_{[t]},$$
$$S_{[t]} = \Theta(H_{[t]} - u_{\text{th}}), \quad U_{[t]} = H_{[t]} - u_{\text{th}} S_{[t]},$$

where $\Theta(\cdot)$ is the Heaviside step function, which outputs 1 if $H_{[t]} > u_{\text{th}}$ and 0 otherwise. For convenience, we denote the temporal spiking process of LIF neuron as $\mathcal{SN}(U)$.

### 3.2. Spike-Driven Video Transformer

The architecture of our proposed SpikeVideoFormer is illustrated in Fig. 1. We adopt a `Conv+ViT` architecture, which has demonstrated scalability and generalization for both ANNs (Yu et al., 2022) and SNNs (Yao et al., 2024a).

Assuming an input video with a shape of $T \times H \times W \times 3$, the data undergoes temporal spiking over time and is then passed through two spike-driven CNN blocks, along with downsample modules, resulting in an output shape of $T \times \frac{H}{8} \times \frac{W}{8} \times 4C$. Each CNN block includes a separable convolution module (Sandler et al., 2018) followed by a channel convolution module, described as follows:

$$U' = U + \text{SepConv}(U),$$
$$\text{SepConv}(U) = \text{Conv}_{\text{pw}}(\text{Conv}_{\text{dw}}(\mathcal{SN}(\text{Conv}_{\text{pw}}(\mathcal{SN}(U))))),$$
$$U'' = U' + \text{ChannelConv}(U'),$$
$$\text{ChannelConv}(U') = \text{Conv}(\mathcal{SN}(\text{Conv}(\mathcal{SN}(U')))).$$
$$(1)$$

| Spike Query | | | | Attention Score | | Spike Query | | | | Attention Score | |
|---|---|---|---|---|---|---|---|---|---|---|---|
| | ⋀ | | | Dot. | Ham. | ⋀ | | | ⋀ | Dot. | Ham. |
| **Spike Keys** | | | | | | **Spike Keys** | | | | | |
| ⋀ | | ⋀ | ⋀ | 0 | 2/6 | | ⋀ | | ⋀ | 1 | 4/6 |
| | | | ⋀ | 0 | 4/6 | | | | ⋀ | 1 | 5/6 |
| ⋀ | | | ⋀ | 0 | 3/6 | ⋀ | | ⋀ | ⋀ | 1 | 3/6 |
| ⋀ | ⋀ | ⋀ | ⋀ | 0 | 1/6 | ⋀ | ⋀ | ⋀ | ⋀ | 1 | 2/6 |
| | | | | ✗ | ✓ | | | | | ✗ | ✓ |

*Figure 2.* Intuitive comparison of attention scores between spike query and keys using Dot-product (*Dot.*) and normalized Hamming similarity (*Ham.*). When the spike query contains no elements, the dot-product ignores the corresponding elements in the spike keys, resulting in identical scores for four distinct spike keys as examples. This illustrates the dot-product's limitation in accurately capturing the similarity between binary spike vectors.

Here, $\text{Conv}_{\text{pw}}$, $\text{Conv}_{\text{dw}}$, and $\text{Conv}$ represent point-wise, depth-wise, and normal convolution (Chollet, 2017), respectively. Temporal dependencies within these blocks are encoded through temporal spiking via neuron-level membrane potential accumulation.

Subsequently, the data is further processed by two spike-driven spatiotemporal transformers, along with downsample modules, resulting in an output shape of $T \times \frac{H}{16} \times \frac{W}{16} \times 10C$. As shown in the lower right corner of Fig. 1, here each spatiotemporal transformer comprises a spike-driven space-time joint attention module followed by a channel MLP module, as described below:

$$
\begin{aligned}
Q_S, K_S, V_S &= \text{Reshape}(\mathcal{SN}(\text{RepConv}(U))), \\
U' &= U + \text{RepConv}(\text{SDHA}(Q_S, K_S, V_S)), \\
U'' &= U' + \text{ChannelMLP}(U'), \\
\text{ChannelMLP}(U') &= \text{MLP}(\mathcal{SN}(\text{MLP}(\mathcal{SN}(U')))).
\end{aligned}
\tag{2}
$$

Here, RepConv refers to re-parametrization convolution (Ding et al., 2021) and the joint attention is based on SDHA. The spike-driven spatiotemporal transformer effectively encodes both spatial and temporal features, enabling robust performance for downstream video tasks. Next, we will elaborate on each key component of SpikeVideoFormer.

### 3.3. Spike-Driven Hamming Attention (SDHA)

Spike-Driven Self-Attention (SDSA) (Zhou et al., 2022; Yao et al., 2024a) uses binary spiking tensors $Q_s, K_s, V_s \in \{0,1\}^{N \times D}$ as the Query, Key and Value, where $N$ is the token length and $D$ is the channel size. By adopting *dot-product* as the attention score and omitting the softmax function, the computation order can be rearranged to achieve linear computational complexity w.r.t. the token length. This process is described as follows:

$$
\text{SDSA} = \mathcal{SN}_s\big(\underbrace{(Q_s K_s^\top)V_s}_{\mathcal{O}(N^2 D)}\big) = \mathcal{SN}_s\big(\underbrace{Q_s(K_s^\top V_s)}_{\mathcal{O}(N D^2)}\big), \tag{3}
$$

where $\mathcal{SN}_s$ denotes LIF neuron with scaled threshold $s \cdot u_{\text{th}}$.

However, the dot-product is not well-suited as the attention score in spike-driven attention. Specifically, *when there are zero elements in the spike query, the dot-product will indiscriminately disregard the corresponding elements in the spike keys*, which can lead to feature vanishing or confusion. To illustrate this issue, we present two intuitive examples in Fig. 2. The dot-product produces identical scores for four significantly different spike keys, indicating that the dot-product used in (Vaswani et al., 2017; Zhou et al., 2022; Yao et al., 2024a) fails to accurately capture the similarity between two binary spike vectors. Additionally, the threshold scale $s$ in Eq. (3) is set empirically, without clear guidance on an appropriate value for stable training. In contrast, we introduce Proposition 3.1 in this paper, which provides theoretical insights for adapting the design of traditional real-valued attention to spike-driven attention.

**Proposition 3.1** (JL Lemma on Binary Embedding (Jacques et al., 2013; Yi et al., 2015))**.** *Let $q, k \in \mathbb{R}^C$ be real-valued query and key vectors. The corresponding binary embeddings $q_s, k_s \in \{0,1\}^D$ are defined as:*

$$
q_s = sign(Aq), \quad k_s = sign(Ak), \tag{4}
$$

*where $A \in \mathbb{R}^{C \times D}$ is a projection matrix with each element drawn independently from a normal distribution $\mathcal{N}(0,1)$. Given any $\delta > 0$ and $D > \frac{\log M}{\delta^2}$, we have:*

$$
P\Big(\big|f_{\mathcal{H}}(q_s, k_s) - g\big(f_{\mathcal{C}}(q,k)\big)\big| \leq \delta\Big) \geq 1 - 2e^{-\delta^2 D}, \tag{5}
$$

*where $P(\cdot)$ denotes probability, $g(x) = 1 - \frac{1}{\pi}\arccos(x)$ is a continuous and monotone function for $x$ with $g(x) \in [0,1]$, and $M$ represents the number of all possible keys and queries in the finite training set. $f_{\mathcal{H}} \in [0,1]$ is the normalized Hamming similarity defined as:*

$$
f_{\mathcal{H}}(q_s, k_s) = 1 - \frac{1}{D}\sum_{i=1}^{D} \mathbb{1}(q_s^{(i)} \neq k_s^{(i)}), \tag{6}
$$

*and $f_{\mathcal{C}} \in [0,1]$ is the cosine similarity defined as:*

$$
f_{\mathcal{C}}(q,k) = \frac{q^\top k}{\|q\|\|k\|}. \tag{7}
$$

The proof is provided in Appendix A. This proposition shows that when the channel size $D$ is sufficiently large, the normalized Hamming similarity $f_{\mathcal{H}}$ closely approximates $g(f_{\mathcal{C}})$ with high probability, where $f_{\mathcal{C}}$ is used in traditional real-valued attention. Empirical evidence supporting this is shown in Fig. 3. We randomly generate 10k, 100k, and 1000k pairs of real-valued unit vectors of size 64, compute their corresponding binary embeddings of size $D$ and evaluate the average error $|f_{\mathcal{H}} - g(f_{\mathcal{C}})|$. The results indicate that as $D$ increases, the error approaches zero. Additionally, larger datasets generally require large $D$ to achieve a close

approximation in similarity measurement. Since $g(\cdot)$ is a continuous and monotonic function, $g(f_\mathcal{C})$ can still preserve the similarity ranking between a query and different keys. Therefore, we propose to use the normalized Hamming similarity $f_\mathcal{H}$ as the attention score function in SDHA.

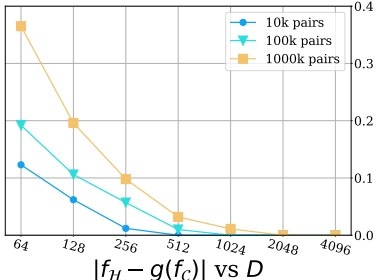

$$|f_\mathcal{H} - g(f_\mathcal{C})| \text{ vs } D$$

*Figure 3.* Empirical evidence of average error $|f_\mathcal{H} - g(f_\mathcal{C})|$ with respect to embeddings size $D$.

**Computational Efficiency Design.** Directly using $f_\mathcal{H}$ for spike-driven attention faces three main challenges: (1) floating-point operations, (2) non-differentiable computation, and (3) non-linear complexity w.r.t. the token length. To address these issues, we rewrite $f_\mathcal{H}$ as:

$$f_\mathcal{H}(q_s, k_s) = \frac{1}{2} + \frac{1}{2D}(2q_s - \mathbf{1})^\top(2k_s - \mathbf{1}), \quad (8)$$

with the derivative provided in Appendix B. Since the constant $\frac{1}{2}$ does not affect the similarity ranking between a query and different keys, we omit it in Eq. (8). Next, by replacing $f_\mathcal{H}$ with the dot-product in Eq. (3), we define the SDHA as:

$$\text{SDHA} = \mathcal{SN}\Big(\Big[\frac{1}{2D}(2Q_s - \mathbf{1})(2K_s - \mathbf{1})^\top\Big]V_s\Big) \quad (9)$$

or equivalently,

$$\text{SDHA} = \mathcal{SN}_{2D}\Big(\underbrace{(2Q_s - \mathbf{1})\Big[(2K_s - \mathbf{1})^\top V_s\Big]}_{\mathcal{O}(ND^2)}\Big). \quad (10)$$

Here, the transformations $(2Q_s - 1)$ and $(2K_s - 1)$ map elements from $\{0, 1\}$ to $\{-1, 1\}$, which can be efficiently achieved through element-wise bit shifts, eliminating the need for multiplication. The scaling factor $\frac{1}{2D}$ can be directly merged into the threshold of spiking neuron, thus avoiding multiplication during inference. After rearranging the computation order, the computational complexity of our proposed SDHA remains linear w.r.t. the token length. Finally, the matrix multiplication in Eq. (10) can be converted to additions and extractions via addressing algorithms (Frenkel et al., 2023).

### 3.4. Space-Time Spike-Driven Attention Designs

Existing spike-driven attention designs are primarily tailored for single-image tasks, focusing solely on spatial feature encoding. However, video-based vision tasks require effective

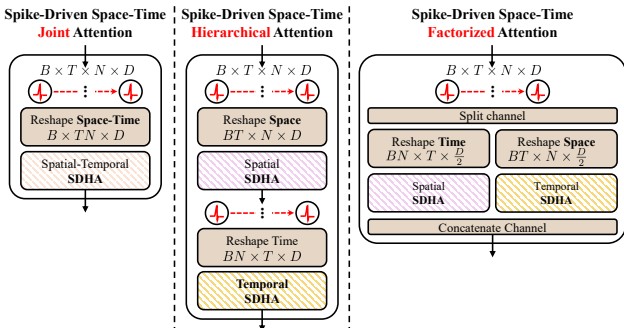

*Figure 4.* Space-time Spike-Driven Attention Designs.

*Table 1.* Comparison of complexity and parameters between ANN-based and spike-driven space-time attention designs.

| | ANN-based Attention | | Spike-Driven Attention | |
|---|---|---|---|---|
| | Complexity | # Params | Complexity | # Params |
| Joint | $\mathcal{O}(T^2N^2D)$ | $4D^2$ | $\mathcal{O}(TND^2)$ | $4D^2$ |
| Hierarchical | $\mathcal{O}(TN(T+N)D)$ | $8D^2$ | $\mathcal{O}(TND^2)$ | $8D^2$ |
| Factorized | $\mathcal{O}(TN(T+N)D)$ | $7D^2$ | $\mathcal{O}(TND^2)$ | $7D^2$ |

designs for spatiotemporal feature encoding. To address this, we adapt three typical attention designs from ANN-based space-time attention (Arnab et al., 2021) to spike-driven scenarios. These designs are illustrated in Fig. 4, with detailed architectures provided in Appendix C.

Assume the input spike feature has the shape $B \times T \times N \times D$, where $B$ is the batch size, $T$ is the temporal token length, $N$ is the spatial token length, and $D$ is the channel size. The three adapted designs are as follows:

- *1) Joint Attention*: The spike feature is reshaped to $B \times TN \times D$, where the total token length $TN$ spans the entire spatiotemporal domain. This reshaped feature is then processed using SDHA for spatiotemporal feature fusion.

- *2) Hierarchical Attention*: The spike feature is first reshaped to $BT \times N \times D$, with $N$ as the token length for spatial attention. The resulting feature is then reshaped to $BN \times T \times D$, with $T$ as the token length for temporal attention.

- *3) Factorized Attention*: The channel dimension of the input spike feature is divided into two branches. One branch undergoes spatial attention, while the other undergoes temporal attention. The final output is obtained by concatenating these two processed parts.

We compare the computational complexity and parameter count of the three prototypes in Tab. 1. The ANN-based space-time joint attention suffers from quadratic complexity w.r.t. spatiotemporal length, $\mathcal{O}(T^2N^2D)$. While the

hierarchical and factorized designs attempt to reduce this to $\mathcal{O}(TN(T+N)D)$, their complexity remains quadratic w.r.t. $T$ and $N$. In contrast, all three spike-driven attention designs achieve linear complexity w.r.t. spatial or temporal length, $\mathcal{O}(TND^2)$, making them significantly more *efficient* and *scalable* for video-based vision tasks. This is particularly advantageous when processing large volumes of video data, as it alleviates substantial computational costs.

Empirical experiments consistently demonstrate the superior performance of the joint attention design over other designs across different video tasks. Therefore, we adopt this approach in our SpikeVideoFormer, which ensures enhanced performance without increasing computational demands. Notably, this observation also aligns with recent findings in video generation (Yang et al., 2024).

## 4. Experiment

We evaluate our proposed SpikeVideoFormer on three representative video-based vision tasks: **1) Video classification**, a high-level task that predicts a single categorical label for a given video clip; **2) Human pose tracking**, which involves fine-grained regression of pose values over time, with an emphasis on maintaining accuracy during long-term tracking; and **3) Video semantic segmentation**, a dense classification task that assigns a category to each pixel in the input frames, highlighting the importance of similar feature encoding and fusion. Additionally, we conduct ablation studies to further verify the effectiveness of our approach.

**Model Variants.** To ensure a fair comparison with previous works, we use two similar model sizes: a small model with $C = 32$, resulting in 15.1M parameters, and a large model with $C = 64$, resulting in 55.4M parameters. The architecture details are provided in Appendix D. Both models are pre-trained on the ImageNet1K dataset and then used for initialization in downstream video tasks. The pre-training process is provided in Appendix F.

### 4.1. Video Classification

**Implementation Details.** The Kinetics-400 dataset (Kay et al., 2017) is a widely-used dataset for human action recognition, which consists of around 240k training videos and 20k validation videos in 400 human action categories. During training, a clip of 32 video frames from each video using a temporal stride of 2 and spatial size of $224 \times 224$ is employed. We follow prior works (Liu et al., 2022) by reporting top-1 and top-5 recognition accuracy. Additional implementation details are provided in Appendix G

**Experiment Results.** As shown in Tab. 2, our method achieves a top-1 accuracy of 79.8%, surpassing the prior Meta-SpikeFormer by 4.3%, with a similar improvement observed in top-5 accuracy. This improvement is pri-

Table 2. **Video Classification Results on Kinetics-400 dataset.** "Spike", "Param", "Step", "Top-1" and "Top-5" denote spike-driven or not, parameter count, temporal spiking step, top-1 and top-5 accuracy. The best result among SNN methods is **bold**, and among ANN methods is underline.

| Methods | Architecture | Spike | Param (M) | Power (mJ) | Top-1↑ (%) | Top-5↑ (%) |
|---|---|---|---|---|---|---|
| ANN | I3D-101 (Wang et al., 2018) | ✗ | 61.8 | 1651.4 | 77.7 | 93.3 |
| | ViViT (Arnab et al., 2021) | ✗ | 310.8 | 6651.6 | 80.6 | 94.7 |
| | Swin-B (Liu et al., 2022) | ✗ | 88.1 | 1297.2 | 80.6 | 94.6 |
| Transformer-Based SNN | Meta-SpikeFormer (Yao et al., 2024a) | ✓ | 55.9 | 396.4 | 75.5 | 90.1 |
| | SpikeVideoFormer (Ours) | ✓ | 55.9 | 412.1 | **79.8** | **94.0** |

marily attributed to our spike-driven Hamming attention, which outperforms the dot-product attention used in Meta-SpikeFormer. Additionally, our SNN model outperforms the ANN-based I3D-101 by 2.1% and closely matches the performance of the recent Swin-B, trailing by only 0.8% in top-1 accuracy while offering a $\times 3$ increase in efficiency. This efficiency gain is even more pronounced when compared to the earlier work ViViT, with our method showing over a $\times 16$ reduction in power consumption.

### 4.2. Human Pose Tracking

We evaluate our method on the task of human pose tracking using either video clips or event streams as input. Notably, the use of efficient SNNs for this challenging task, particularly in long-term pose tracking, has been largely unexplored. Thus we investigate the application of SNNs to human pose tracking in this work.

**Implementation Details.** The MMHPSD dataset (Zou et al., 2021) is currently the largest dataset offering both videos and event streams, along with annotated 3D poses. We evaluate our method using both modalities as input and conduct a comprehensive comparison with ANN-based and SNN-based approaches for this fine-grained tracking task. Following the standard train-test split of the dataset, we train and evaluate models with $T = 8$ and $T = 32$ time steps, which correspond to 2 and 8 seconds of input video clips or event streams, respectively. The pipeline and additional implementation details are provided in the Appendix H.

**Evaluation metrics.** Following prior works (Kocabas et al., 2020), we report Mean Per Joint Position Error (MPJPE), PELvis-aligned MPJPE (PEL-MPJPE), and Procrustes-Aligned MPJPE (PA-MPJPE) in millimeters (**mm**). PA-MPJPE aligns the predicted and target poses through both rotation and translation, while PEL-MPJPE only aligns them using translation of the root joint.

*Table 3.* **Human Pose Tracking Results on MMHPSD Dataset.** The best result among SNN methods is **bold**, and among ANN methods is underline. *For a fair comparison, we extend the spike-driven spatial attention used in previous image-based SNN methods with the space-time joint attention, while retaining the dot-product as the attention score function. This modification is consistently applied in the subsequent experiments.

| Methods | Architecture | $T=8$ | | | | | $T=32$ | | | |
|---|---|---|---|---|---|---|---|---|---|---|
| | | Param (M) | Power (mJ) | MPJPE↓ | PEL-MPJPE↓ | PA-MPJPE↓ | Power (mJ) | MPJPE↓ | PEL-MPJPE↓ | PA-MPJPE↓ |
| **Video as Input** | | | | | | | | | | |
| ANN | VIBE (Kocabas et al., 2020) | 48.3 | 392.1 | - | 68.2 | 46.3 | 1511.2 | - | 75.4 | 53.6 |
| | GLoT (Shen et al., 2023) | 40.5 | 487.5 | - | 61.6 | 39.9 | 4046.1 | - | 67.3 | 46.5 |
| ANN2SNN | VIBE (Hu et al., 2023) | 48.3 | 87.4 | - | 78.1 | 55.8 | 123.8 | - | 84.3 | 62.2 |
| CNN-Based SNN | SEW-Res (Fang et al., 2021) | 60.6 | 26.4 | 124.7 | 68.7 | 48.8 | 106.7 | 150.1 | 77.4 | 56.8 |
| | MA-SNN (Yao et al., 2023) | 78.8 | 15.7 | 122.4 | 68.2 | 48.2 | 63.4 | 145.3 | 76.8 | 56.0 |
| Transformer-Based SNN | SpikFormer* (Zhou et al., 2022) | 29.9 | 24.5 | 125.2 | 69.7 | 48.8 | 99.6 | 147.6 | 77.8 | 56.9 |
| | | 66.7 | 43.9 | 119.2 | 67.4 | 46.2 | 178.4 | 140.1 | 76.3 | 55.2 |
| | Meta-SpikeFormer* (Yao et al., 2024a) | 15.4 | 34.2 | 126.4 | 71.3 | 49.1 | 138.8 | 145.7 | 77.9 | 57.1 |
| | | 55.8 | 95.7 | 116.3 | 67.0 | 45.7 | 387.2 | 138.2 | 74.2 | 54.5 |
| | SpikeVideoFormer (Ours) | 15.5 | 34.6 | 121.8 | 67.9 | 44.4 | 143.4 | 141.3 | 75.2 | 49.9 |
| | | 55.8 | 96.0 | **109.2** | **61.8** | **39.8** | 391.2 | **131.8** | **69.2** | **47.5** |
| **Event Stream as Input** | | | | | | | | | | |
| ANN | VIBE (Kocabas et al., 2020) | 48.3 | 392.1 | - | 70.9 | 46.6 | 1511.2 | - | 76.8 | 53.9 |
| | GLoT (Shen et al., 2023) | 40.5 | 487.5 | - | 62.0 | 40.1 | 4046.1 | - | 68.8 | 47.7 |
| ANN2SNN | VIBE (Hu et al., 2023) | 48.3 | 84.1 | - | 79.3 | 56.3 | 120.8 | - | 85.8 | 63.6 |
| CNN-Based SNN | SEW-Res (Fang et al., 2021) | 60.2 | 21.5 | 127.3 | 69.7 | 49.1 | 88.7 | 152.3 | 78.0 | 57.1 |
| | MA-SNN (Yao et al., 2023) | 78.4 | 12.1 | 126.7 | 69.4 | 48.9 | 51.3 | 150.2 | 77.7 | 56.7 |
| Transformer-Based SNN | SpikFormer* (Zhou et al., 2022) | 29.9 | 19.7 | 126.1 | 70.1 | 49.0 | 81.2 | 149.1 | 78.4 | 57.6 |
| | | 66.6 | 34.9 | 121.0 | 67.9 | 47.1 | 143.6 | 143.4 | 77.2 | 55.8 |
| | Meta-SpikeFormer* (Yao et al., 2024a) | 15.4 | 27.1 | 130.1 | 72.2 | 49.8 | 112.1 | 148.5 | 78.2 | 57.4 |
| | | 55.8 | 72.5 | 117.1 | 67.5 | 46.0 | 308.5 | 140.8 | 75.0 | 55.1 |
| | SpikeVideoFormer (Ours) | 15.5 | 27.6 | 128.9 | 68.3 | 46.9 | 121.8 | 142.5 | 75.9 | 50.3 |
| | | 55.8 | 73.3 | **115.7** | **62.1** | **40.3** | 291.3 | **136.7** | **70.1** | **48.1** |

**Experiment Results.** In Tab. 3, we compare SpikeVideoFormer with four SNN-based models using either video clips or event streams as input. Our method outperforms prior SNN methods by a significant margin, achieving a PA-MPJPE of 39.8, compared to 48.2 for CNN-based SNN and 45.7 for Transformer-based SNN. This demonstrates that, unlike CNN-based SNN, the spike-driven transformer excels at encoding global features for video-based tasks. Additionally, our proposed Hamming attention improves similarity measurement over spike-driven dot-product attention used in SpikFormer and Meta-SpikFormer, while not sacrificing computational efficiency.

When compared to two representative ANN-based methods, our performance is close to that of GLoT, with a minor gap of 0.2mm in PEL-MPJPE and 0.1mm in PA-MPJPE. However, for long-term tracking ($T=32$), the gap increases to 1.9mm and 1.0mm, respectively. Despite this, the power consumption of our method increases from 96.0 to 391.2 (a $4.1\times$ increase), whereas GLoT's power consumption rises from 487.5 to 4046.1 (an $8.3\times$ increase). This highlights the

computational efficiency of SpikeVideoFormer, which has $\mathcal{O}(T)$ complexity. Furthermore, when using event streams as input, spike-driven methods exhibit lower power consumption compared to video clips. This is likely due to the sparsity of events, which typically results in a lower spiking rate in SNN models. In contrast, ANN models still require the same level of computation regardless of whether the input is an event stream or a video clip.

**Inference Time Discussion.** We test on an A6000 GPU and AMD EPYC 7543 CPU for human pose tracking, averaged over 1,000 video clips with a batch size of 1. As $T$ increases from 8 to 32 (4x), GLoT (quadratic ANN attention) achieves the best performance but experiences a 9.8× increase in inference time from 303 to 2972ms, while VIBE (linear ANN-GRU) shows a 5.1x increase from 264 to 1335ms but performs the worst. Our SNN method exhibits only 4.6× increase from 235 to 1087ms thanks to our proposed spiking space-time linear attention.

For edge device deployment, SNNs, relying only on additions, significantly reduce power consumption and latency

on neuron-computing devices (Kudithipudi et al., 2025; Yao et al., 2025). ANNs, requiring floating-point multiplications, achieve high parallel acceleration on GPUs but at a high energy cost. On the AMD Xilinx ZCU104 (Li et al., 2023a), ANNs process at 691.2 GFLOPs/s, while SNNs reach 5529.6 GFLOPs/s—an 8x speedup. In 45nm technology (Horowitz, 2014), ANN multiplication consumes 4.6pJ, whereas SNN addition uses only 0.9pJ, a 5.1x energy reduction.

### 4.3. Video Semantic Segmentation (VSS)

We further evaluate our proposed SpikeVideoFormer on the task of VSS. Since our primary objective is to demonstrate the generalization ability of our method across various video-based tasks, we avoid using a complex architecture for VSS. Specifically, the classification head in SpikeVideoFormer consists of two main components: 1) Memory Read and Fusion modules (Paul et al., 2021), where relevant semantic information from previous frames (memory) is read and fused with the features of the current frame via spike-driven cross-attention, and 2) Spike-Driven Feature Pyramid Networks (SpikeFPN) (Zhao et al., 2017), which aggregate multi-scale features extracted by SpikeVideoFormer to predict pixel-level semantic categories. The pipeline is detailed in Appendix I. Unlike recent SNN methods (Su et al., 2024; Wang et al., 2024) that focus on single-frame semantic segmentation, our work explores the use of efficient SNNs for VSS, advancing their application to spatiotemporal contexts.

**Datasets.** We conduct experiments on two widely used VSS datasets: CityScapes (Cordts et al., 2016) and VSPW (Miao et al., 2021). CityScapes focuses on driving scenarios and includes 3,000 high-resolution images with fine annotations for training and 500 for validation, covering 19 semantic categories. In contrast, VSPW is the largest VSS benchmark, encompassing a wide range of indoor and outdoor scenes with annotations across 124 categories. This dataset consists of 2,806 clips for training and 343 clips for validation.

**Implementation Details.** Following prior work (Sun et al., 2024), we evaluate a small-size model (17.8M) on the CityScapes dataset and a large-size model (72.1M) on the VSPW dataset. We also employ the Integer-LIF model (Luo et al., 2024) in SpikeFPN, which has been shown to improve performance with minimal computational increase. A setting of 1 means the output spikes are $\{0, 1\}$, and a setting of 4 means $\{0, 1, 2, 3\}$. We begin by training the encoder and decoder for single-frame segmentation, with the encoder initialized using the pre-trained model from Appendix F. Once trained, we fix the encoder and decoder, and incorporate the Memory Read and Fusion module, which are trained on 4-frame clips for VSS. We report the performance of VSS using the mean Intersection over Union (mIoU). The power consumption is averaged over the number of frames. Addi-

*Table 4.* **VSS Results on the CityScape dataset.** "Integer-LIF" refers to the use of the integer LIF model (Luo et al., 2024) in the SpikeFPN segmentation head.

| Methods | Architecture | Integer -LIF | Param (M) | Power (mJ) | mIoU↑ (%) |
|---|---|---|---|---|---|
| ANN | FCN (Long et al., 2017) | - | 9.8 | 534.8 | 61.5 |
| | PSPNet (Zhao et al., 2017) | - | 13.7 | 793.3 | 70.2 |
| | SegFormer (Xie et al., 2021) | - | 13.8 | 270.2 | 74.1 |
| | CFFM (Sun et al., 2024) | - | 15.4 | 376.5 | 75.1 |
| Transformer -Based SNN | SpikFormer (Zhou et al., 2022) | 1 | 35.8 | 42.3 | 62.3 |
| | | 4 | 35.8 | 48.9 | 65.0 |
| | Meta-SpikeFormer (Yao et al., 2024a) | 1 | 17.8 | 61.6 | 63.0 |
| | | 4 | 17.8 | 63.5 | 65.9 |
| | SpikeVideoFormer (Ours) | 1 | 17.8 | 64.0 | 70.8 |
| | | 4 | 17.8 | 65.3 | **73.1** |

*Table 5.* **VSS Results on the VSPW dataset.** "Integer-LIF" refers to the use of the integer LIF model (Luo et al., 2024) in the SpikeFPN segmentation head.

| Methods | Architecture | Integer -LIF | Param (M) | Power (mJ) | mIoU↑ (%) |
|---|---|---|---|---|---|
| ANN | DeepLab (Chen et al., 2017) | - | 62.7 | - | 34.7 |
| | PSPNet (Zhao et al., 2017) | - | 70.5 | 1313.9 | 36.5 |
| | TCB (Miao et al., 2021) | - | 70.5 | 1206.9 | 37.5 |
| | CFFM (Sun et al., 2024) | - | 85.5 | 1068.1 | 49.3 |
| Transformer -Based SNN | SpikFormer (Zhou et al., 2022) | 1 | 92.2 | 176.8 | 30.0 |
| | | 4 | 92.2 | 180.3 | 31.0 |
| | Meta-SpikeFormer (Yao et al., 2024a) | 1 | 72.1 | 208.7 | 31.1 |
| | | 4 | 72.1 | 213.1 | 32.3 |
| | SpikeVideoFormer (Ours) | 1 | 72.1 | 218.8 | 36.5 |
| | | 4 | 72.1 | 226.2 | **37.9** |

tional implementation details are provided in Appendix I.

**Experiment Results.** From Tab. 4, we observe that our method outperforms classical ANN approaches, such as FCN and PSPNet, achieving a higher mIoU of 70.8, compared to 61.5 and 70.2, while delivering ×12 and ×8 greater efficiency, respectively. When compared to more complex models like SegFormer and CFFM, SpikeVideoFormer with Integer-LIF of 4 achieves competitive mIoU scores, trailing by only 1% and 2%, while offering a ×4 boost in efficiency. These results underscore the efficient yet effective space-time feature encoding capability of our SNN model. Furthermore, when compared to recent SNN approaches, SpikeVideoFormer achieves a significantly higher mIoU (73.1 vs. 65.0 for SpikFormer and 65.9 for Meta-SpikeFormer). This performance gap is likely due to the limitations of the dot-product mechanism in these models, which struggles to accurately capture feature similarity during space-time feature fusion in the backbone and memory read module.

A similar trend is observed in Tab. 5 on the VSPW dataset, where our method outperforms classical ANN approaches

*Table 6.* **Ablation Study.** We perform experiments on two tasks: human pose tracking using the MMHPSD dataset and video semantic segmentation using the Cityscapes dataset. In our model, the value of $D$ varies across layers rather than being fixed.

| Architecture | | Power (mJ) | Pose-Track PA-MPJPE↓ | Power (mJ) | VSS mIoU(%)↑ |
|---|---|---|---|---|---|
| SpikeVideoFormer | | 96.0 | **39.8** | 65.3 | **73.1** |
| Att. Score | Ham → Dot-prod | 95.7 | 45.7 (+5.9) | 63.5 | 65.9 (-7.8) |
| Space-Time Attention | Joint → Hierarchical | 102.6 | 44.5 (+4.7) | 68.5 | 71.7 (-1.4) |
| | Joint → Factorized | 99.7 | 45.1 (+5.3) | 67.2 | 70.3 (-2.8) |
| | Joint → Neuron-Level | 94.6 | 47.4 (+7.6) | 65.1 | 67.2 (-5.9) |
| | Joint → Spatial-Only | 90.2 | 54.2 (+14.4) | 64.2 | 62.1 (-11.0) |
| Initial | Pre-train → Random | 96.5 | 53.8 (+14.0) | 65.3 | 61.3 (-11.8) |
| Threshold Scale $s$ in Eq. (10) | $s = 1/2D \rightarrow 1/8$ | 97.8 | 46.7 (+6.9) | 66.7 | 68.5 (-4.6) |
| | $s = 1/2D \rightarrow 1/64$ | 97.0 | 43.0 (+3.2) | 66.0 | 71.1 (-2.0) |
| | $s = 1/2D \rightarrow 1/512$ | 96.3 | 41.3 (+1.5) | 63.2 | 70.3 (-2.8) |
| | $s = 1/2D \rightarrow 1/2048$ | 91.4 | 48.3 (+8.5) | 60.3 | 67.4 (-5.7) |

DeepLab, PSPNet, and TCB. While there is a notable performance gap compared to the latest CFFM model, this can be attributed to CFFM's use of a complex coarse-to-fine feature assembly module for memory read and fusion. In contrast, our scope is to demonstrate the capability and generalization ability of our SNN model as a backbone for video-based tasks, using only a simple architecture of head for VSS.

### 4.4. Ablation Study

We conduct ablation studies on various components of SpikeVideoFormer on both tasks in Tab. 6.

**Attention Score Function.** Performance drops significantly in both tasks when replacing our proposed Hamming attention with dot-product attention, 15% in pose tracking and 10% in VSS. The slight reduction in power may be attributed to the dot-product's masking effect, which lowers the spiking rate after applying attention.

**Space-Time Attention.** Replacing joint space-time attention with hierarchical or factorized attention results in performance degradation in both tasks, primarily due to the failure to capture global spatiotemporal features. This aligns with recent findings in video generation (Yang et al., 2024), where joint attention outperforms other decomposed attentions. When only neuron-level temporal encoding is used (Neuron-Only) without attention on temporal domain, performance drops moderately but remains acceptable. However, removing neuron-level temporal encoding (Spatial-Only) by resetting the membrane potential at each time step causes a significant decline, highlighting the capability of temporal encoding inherent to SNNs.

**Initialization.** Initializing the model with pre-trained weights from the ImageNet1K dataset significantly boosts performance on both tasks, improving PA-MPJPE by over 26% and mIoU by 19%.

**Threshold Scale $s$ in Eq. (10).** Prior work (Yao et al., 2024a) sets $s = 1/8$ in Eq. (3). However, as shown in Tab. 6, we observe that fixed values of $s$ generally result in worse performance compared to the proposed $s = 1/2D$, where $D$ varies across layers in our model.

## 5. Conclusion

In this paper, we present SpikeVideoFormer, an efficient spike-driven video Transformer designed to enhance the performance of SNNs in video-based vision tasks. Our model leverages spike-driven Hamming attention and joint space-time attention to achieve state-of-the-art results among SNN approaches for video classification, human pose tracking, and video semantic segmentation, while significantly improving efficiency compared to recent ANN-based methods. Additionally, the linear temporal complexity of $\mathcal{O}(T)$ further highlights the potential of our proposed SNN-based Transformer for scalable video processing. In the future work, we aim to scale up the SNN model as a backbone to support a broader range of video tasks, such as video understanding and generation.

## Acknowledgments

This work was partially supported by a grant from Shenzhen Science and Technology Program (Grant No. JSGG20220831105002004).

## Impact Statement

This paper presents work whose goal is to advance the field of Machine Learning. There are many potential societal consequences of our work, none which we feel must be specifically highlighted here.

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

## A. Proof of Proposition 3.1

**Lemma A.1** (Johnson–Lindenstrauss Lemma on Binary Embedding (Jacques et al., 2013; Yi et al., 2015)). *Let $\{x_i\}_{i=1}^M$ be set of $M$ real-valued points, define its one bit quantization of the projections,*

$$b(x) = sign(Ax), \tag{11}$$

*where $b(x) \in \{0,1\}^D$ is the binary embedding of $x \in \mathbb{R}^C$ and $A \in \mathbb{R}^{D \times C}$ is a projection matrix with each entry generated independently from the normal distribution $\mathcal{N}(0,1)$. Given $D > \frac{\log M}{\delta^2}$, for any two points among $M$, we have*

$$P\Big(|d_{\mathcal{H}}(b_i, b_j) - d_{\mathcal{C}}(x_i, x_j)| \le \delta\Big) \ge 1 - 2e^{-\delta^2 D}. \tag{12}$$

*Here $d_{\mathcal{H}}$ is the normalized Hamming distance defined as*

$$d_{\mathcal{H}}(b_i, b_j) = \frac{1}{C_k} \sum_{k=1}^D \mathbb{1}(b_{ik} \ne b_{jk}), \tag{13}$$

*and $d_{\mathcal{C}}$ is cosine distance defined as*

$$d_{\mathcal{C}}(x_i, x_j) = \frac{1}{\pi} \arccos\Big(\frac{x_i^\top x_j}{\|x_i\|\|x_j\|}\Big). \tag{14}$$

According to Lemma A.1, we provide the proof of Proposition 3.1 as follows:

*Proof.* By substituting $x_i, x_j$ and $b_i, b_j$ in Eq. (12) with $q, k$ and $q_s, k_s$ respectively, we obtain:

$$|d_{\mathcal{H}}(q_s, k_s) - d_{\mathcal{C}}(q, k)| \le \delta. \tag{15}$$

Next, using the definitions from Eq. (6) and (7), we have:

$$d_{\mathcal{H}}(q_s, k_s) = 1 - f_{\mathcal{H}}(q_s, k_s), \tag{16}$$
$$d_{\mathcal{C}}(q, k) = 1 - g\big(f_{\mathcal{C}}(q, k)\big), \tag{17}$$

where $g(x) = 1 - \frac{1}{\pi} \arccos(x)$. Substituting these into the inequality, we get:

$$\Big|\big(1 - f_{\mathcal{H}}(q_s, k_s)\big) - \big(1 - g\big(f_{\mathcal{C}}(q, k)\big)\big)\Big| \le \delta, \tag{18}$$

which simplifies to:

$$\Big|f_{\mathcal{H}}(q_s, k_s) - g\big(f_{\mathcal{C}}(q, k)\big)\Big| \le \delta. \tag{19}$$

Finally, substituting this result into Eq. (12), we obtain:

$$P\Big(|f_{\mathcal{H}}(q_s, k_s) - g\big(f_{\mathcal{C}}(q, k)\big)| \le \delta\Big) \ge 1 - 2e^{-\delta^2 D}. \tag{20}$$

$\square$

## B. Derivative of Normalized Hamming Similarity

We define $\mathbf{1} \in \{1\}^{1 \times D}$ as all-one vector. For the spike vectors $q_s, k_s \in \{0,1\}^{1 \times D}$, we rewrite the normalized Hamming similarity defined in Eq. (6) as:

$$
\begin{aligned}
f_{\mathcal{H}}(q_s, k_s) &= 1 - \frac{1}{D} \sum_{i=1}^D \mathbb{1}(q_s^{(i)} \ne k_s^{(i)}) \\
&= 1 - \frac{1}{D}\Big(q_s^\top(\mathbf{1} - k_s) + (\mathbf{1} - q_s)^\top k_s\Big) \\
&= 1 - \frac{1}{D}\Big(q_s^\top \mathbf{1} + \mathbf{1}^\top k_s - 2q_s^\top k_s\Big) \\
&= 1 - \frac{1}{D}\Big(q_s^\top \mathbf{1} + \mathbf{1}^\top k_s - 2q_s^\top k_s - \frac{D}{2} + \frac{D}{2}\Big) \\
&= 1 - \frac{1}{D}\Big(q_s^\top \mathbf{1} + \mathbf{1}^\top k_s - 2q_s^\top k_s - \frac{D}{2}\Big) - \frac{1}{2} \\
&= \frac{1}{2} + \frac{1}{2D}\Big(4q_s^\top k_s - 2q_s^\top \mathbf{1} - 2\mathbf{1}^\top k_s + D\Big) \\
&= \frac{1}{2} + \frac{1}{2D}\big(2q_s - \mathbf{1}\big)^\top\big(2k_s - \mathbf{1}\big). \tag{21}
\end{aligned}
$$

Therefore, for the spike query and key matrices, denoted as $Q_s, K_s \in \{0,1\}^{N \times D}$, and the all-ones matrix $\mathbf{1} \in \{1\}^{N \times D}$, we have the corresponding attention score expression as:

$$f_{\mathcal{H}}(Q_s, K_s) = \frac{1}{2} + \frac{1}{2D}\big(2Q_s - \mathbf{1}\big)\big(2K_s - \mathbf{1}\big)^\top. \tag{22}$$

After omitting the constant $\frac{1}{2}$, we replace the dot-product similarity used in Eq. (3) with normalized Hamming similarity:

$$
\begin{aligned}
\text{SDHA} &= \mathcal{SN}_{2D}\Big(\underbrace{\Big[\frac{1}{2D}(2Q_s - \mathbf{1})(2K_s - \mathbf{1})^\top\Big]V_s}_{\mathcal{O}(N^2 D)}\Big) \\
&= \mathcal{SN}_{2D}\Big(\underbrace{(2Q_s - \mathbf{1})\big[(2K_s - \mathbf{1})^\top V_s\big]}_{\mathcal{O}(ND^2)}\Big). \tag{23}
\end{aligned}
$$

## C. Spike-Driven Space-Time Attention

The detailed architecture of three different spike-driven space-time attentions is shown in Fig. 5. For simplicity, we assume the hidden channel size of attention module is identical with the input channel size $D$ and the number of attention head is $M$. The detailed description of architecture and FLOPs calculation [2] for three prototypes are presented as follows:

- *Spike-Driven Space-Time Joint Attention*: The input spike feature goes through three parallel linear, batch

---

[2] When calculating FLOPs, we assume the batch size $B$ is 1 for simplicity.

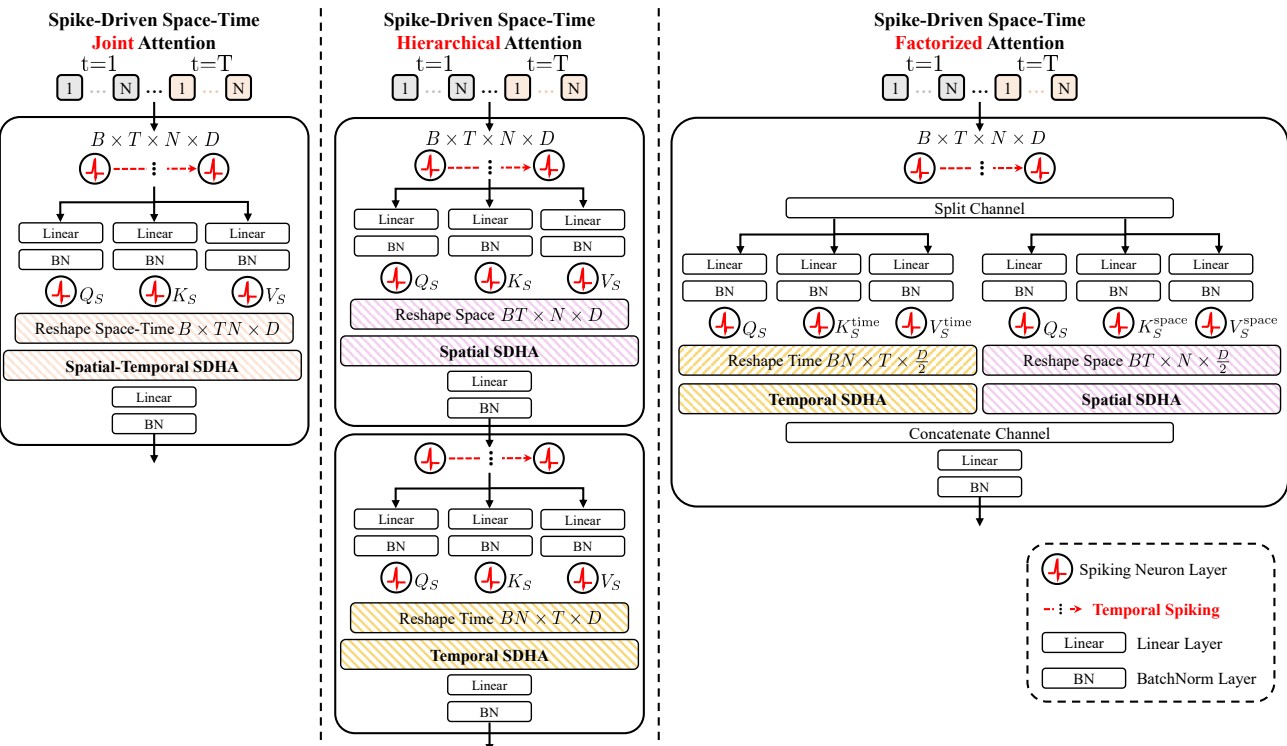

*Figure 5.* Architecture Details of three different spike-driven space-time attentions.

normalization and spiking layers to convert to spike query $Q_S$, key $K_S$ and value $V_S$. They will be reshaped to $B \times TN \times D$ and forwarded to SDHA for spatiotemporal feature fusion. The output finally goes through a linear and batch normalization layer. In total, the number of parameters are $4D^2$.

$$\text{FLOPs} = \underbrace{2TN(\frac{D}{M})^2 M}_{\text{Ham. Att.}} + \underbrace{4TN(D^2 + D)}_{\text{Linear+BN}} \tag{24}$$
$$= \mathcal{O}(TND^2),$$

- *Spike-Driven Space-Time Hierarchical Attention*: The input spike feature goes through three parallel linear, batch normalization and spiking layers to convert to spike query $Q_S$, key $K_S$ and value $V_S$. They will be reshaped to $BT \times N \times D$ and forwarded to SDHA for spatial feature fusion. The output then goes through a linear and batch normalization layer. This process is repeated while the spike feature is reshaped to $BN \times T \times D$ for temporal feature fusion. In total, the number of parameters are $8D^2$.

$$\text{FLOPs} = \underbrace{2TN(\frac{D}{M})^2 M + 4TN(D^2 + D)}_{\text{Spatial Ham. Att.}}$$
$$+ \underbrace{2NT(\frac{D}{M})^2 M + 4NT(D^2 + D)}_{\text{Temporal Ham. Att.}} \tag{25}$$
$$= \mathcal{O}(TND^2),$$

- *Spike-Driven Space-Time Factorized Attention*: The input spike feature will be first split into two branches with half of the channel size $\frac{D}{2}$. The split features will go through five parallel linear, batch normalization and spiking layers to convert to spike query $Q_S$, key $K_S^{\text{time}}, K_S^{\text{space}}$ and value $V_S^{\text{time}}, V_S^{\text{space}}$. The query $Q_S$ is reused for the following temporal and spatial hamming attention. The output is the concatenation of two branches and then goes through a linear and batch normalization layer. In total, the number of parameters are $7D^2$.

$$\text{FLOPs} = \underbrace{2TN(\frac{D/2}{M})^2 M + 3TN((\frac{D}{2})^2 + \frac{D}{2})}_{\text{Temporal Ham. Att.}}$$
$$+ \underbrace{2NT(\frac{D/2}{M})^2 M + 2TN((\frac{D}{2})^2 + \frac{D}{2})}_{\text{Spatial Ham. Att.}} \tag{26}$$
$$= \mathcal{O}(TND^2),$$

## D. Architecture Details

The detailed model architecture is presented in Tab. 7.

## E. Power Consumption

The computational and energy consumption of Spiking Neural Networks (SNNs) are typically lower than that of Ar-

*Table 7.* Architecture Details of SpikeVideoFormer. For simplicity, we omit the temporal spiking layer here.

| Blocks | Layers | | Shape | Channel (15M) | Channel (55M) |
|---|---|---|---|---|---|
| 1 | DownSample×1 | Conv(kernel=7x7, stride=2) | $T \times \frac{H}{2} \times \frac{W}{2} \times C$ | 32 | 64 |
| | Spik-Driven CNN Block×1 | SepConv(kernel=7x7, stride=1) ChannelConv(kernel=3x3, stride=1) | | | |
| | DownSample×1 | Conv(kernel=3x3, stride=2) | $T \times \frac{H}{4} \times \frac{W}{4} \times 2C$ | 64 | 128 |
| | Spik-Driven CNN Block×1 | SepConv(kernel=7x7, stride=1) ChannelConv(kernel=3x3, stride=1) | | | |
| 2 | DownSample×1 | Conv(kernel=3x3, stride=2) | $T \times \frac{H}{8} \times \frac{W}{8} \times 4C$ | 128 | 256 |
| | Spik-Driven CNN Block×2 | SepConv(kernel=7x7, stride=1) ChannelConv(kernel=3x3, stride=1) | | | |
| 3 | DownSample×1 | Conv(kernel=3x3, stride=2) | $T \times \frac{H}{16} \times \frac{W}{16} \times 8C$ | 256 | 512 |
| | Spike-Driven Spatiotemporal Transformer×6 | SDHA(channel=8C) ChannelMLP(ratio=4) | | | |
| 3 | DownSample×1 | Conv(kernel=3x3, stride=2) | $T \times \frac{H}{16} \times \frac{W}{16} \times 10C$ | 320 | 640 |
| | Spike-Driven Spatiotemporal Transformer×2 | SDHA(channel=10C) ChannelMLP(ratio=4) | | | |

tificial Neural Networks (ANNs). Following the analysis in (Zhou et al., 2022; Yao et al., 2024a), the energy cost of ANNs is measured in multiply-and-accumulate (MAC) operations. Given the number of floating-point operations (FLOPs) in a layer, the energy cost for ANNs can be expressed as:

$$E_{\text{ANNs}} = \text{FLOPs} * E_{\text{MAC}}, \quad (27)$$

where $E_{\text{MAC}}$ is the energy consumed per MAC operation. In contrast, SNNs only involve additions and extractions, which are less energy-intensive. The energy cost for SNNs is measured in accumulate (AC) operations, and computations involving non-spiking neurons in the preceding layer can be skipped. Thus, the energy cost for SNNs is given by:

$$E_{\text{SNNs}} = \rho * \text{FLOPs} * E_{\text{MAC}}, \quad (28)$$

where $\rho$ is the spiking rate. For reference, a 32-bit floating-point operation in 45nm technology consumes $E_{\text{MAC}} = 4.6pJ$ for MAC and $E_{\text{AC}} = 0.9pJ$ for AC operations (Horowitz, 2014).

## F. ImageNet Pre-training

**Implementation details.** ImageNet-1K (Deng et al., 2009) is a widely used benchmark for vision tasks, containing 1.3M training and 50K validation images across 1K classes. Since this is a single-image task without temporal modeling, the primary objective is to pre-train the backbone model for the downstream video-based vision tasks. To maintain scalability and generalization with previous work, Meta-SpikeFormer, we adopt the same model architecture settings, utilizing channel sizes $C$ of 32 and 64, which result in 15.1M and 55.4M parameters, respectively. The input image size is $224 \times 224$. The model is first trained with a time-step $T = 1$ for 200 epochs on 8 NVIDIA A6000 GPUs, followed

*Table 8.* **Results on ImageNet-1K dataset.** All the SNN-based methods are trained directly without distillation. "Spike", "Param", "Step", and "Acc" denote Spike-Driven, the number of parameters, spiking time-step and accuracy. The best result among the SNN methods are in **bold**.

| Methods | Architecture | Spike | Param (M) | Power (mJ) | Step $T$ | Acc↑ (%) |
|---|---|---|---|---|---|---|
| ANN | RSB (Wightman et al., 2021) | ✗ | 60 | 53.4 | 1 | 81.8 |
| | ViT (Dosovitskiy et al., 2021) | ✗ | 86 | 81.0 | 1 | 79.7 |
| ANN2SNN | VGG-16 (Hu et al., 2023) | ✓ | 138.4 | 44.9 | 7 | 73.0 |
| CNN-Based SNN | SEW-Res (Fang et al., 2021) | ✗ | 60.2 | 12.9 | 4 | 69.2 |
| | MA-SNN (Yao et al., 2023) | ✗ | 78.4 | 7.3 | 4 | 76.3 |
| | MS-Res-SNN (Hu et al., 2024) | ✓ | 77.3 | 10.2 | 4 | 75.3 |
| Transformer-Based SNN | SpikFormer (Zhou et al., 2022) | ✗ | 29.7 | 11.6 | 4 | 73.4 |
| | | ✗ | 66.3 | 21.5 | 4 | 74.8 |
| | Meta-SpikeFormer (Yao et al., 2024a) | ✓ | 15.1 | 4.0 | 1 | 71.5 |
| | | ✓ | 15.1 | 16.7 | 4 | 73.2 |
| | | ✓ | 55.4 | 13.0 | 1 | 78.0 |
| | | ✓ | 55.4 | 52.4 | 4 | 79.7 |
| | SpikeVideoFormer (Ours) | ✓ | 15.1 | 4.0 | 1 | 71.5 |
| | | ✓ | 15.1 | 16.8 | 4 | 73.6 |
| | | ✓ | 55.4 | 13.1 | 1 | **78.5** |
| | | ✓ | 55.4 | 52.6 | 4 | **79.9** |

by fine-tuning with a time-step $T = 4$ for an additional 20 epochs. All other training configurations remain consistent with those in (Yao et al., 2024a).

**Experiment Results.** In Tab. 8, we compare SpikeVideo-Former with previous methods in terms of parameter count, power consumption, and accuracy. Our method achieves 73.6% and 79.9% accuracy for the small (15.1M) and large (55.4M) model sizes, respectively, surpassing the state-of-the-art Meta-SpikeFormer by 0.4% and 0.2%. Notably, the major distinction between these two methods lies in the

*Table 9.* **Results on ImageNet-C dataset.**

| Methods | Noise Gaussian | Shot | Impulse | Blur Defocus | Glass | Motion | Zoom | Weather Snow | Frost | Fog | Bright | Digital Contrast | Elastic | Pixelate | JPEG |
|---|---|---|---|---|---|---|---|---|---|---|---|---|---|---|---|
| ResNet50 | 29.6 | 27.9 | 24.5 | 38.8 | 26.1 | 37.9 | 34.5 | 30.1 | 36.7 | 43.6 | 66.8 | 38.8 | 44.8 | 47.0 | 55.1 |
| VOneResNet50 | 34.6 | 33.4 | 31.9 | 37.8 | 35.7 | 37.4 | 34.0 | 25.2 | 36.8 | 30.1 | 62.4 | 28.5 | 48.7 | 63.3 | 61.0 |
| Ours | 33.1 | 34.8 | 38.0 | 39.1 | 38.9 | 38.1 | 35.9 | 28.3 | 37.0 | 42.1 | 66.2 | 34.5 | 49.3 | 61.4 | 61.8 |

use of spike-driven Hamming attention in our design versus dot-product attention, underscoring the effectiveness of our proposed approach. Additionally, our directly trained SNN model outperforms the ANN-based ViT (Dosovitskiy et al., 2021), achieving 79.9% accuracy compared to 79.7%, marking the first instance of an SNN model surpassing an Transformer-based ANN model in this benchmark.

We also evaluate our approach on the ImageNet-C dataset to assess object recognition performance in noisy environments. We compare our method with VOneResNet50 (Dapello et al., 2020) and ResNet50 (He et al., 2016) in Tab. 9.

# G. Video Classification

For the Kinetics-400 dataset, we train our model for 30 epochs using the AdamW optimizer with a batch size of 32 on 8 NVIDIA A6000 GPUs. The initial learning rates are set to 6e-5 for the pre-trained backbone and 6e-4 for the randomly initialized calssification head. The learning rate is adjusted by a CosineAnnealingLR schedule with a maximum of 35 epochs. For all model variants, we sample 32 frames per video clip using a temporal frame stride of 2, and set the spatial resolution to $224 \times 224$. During inference, we adopt the $4 \times 3$ view strategy from (Liu et al., 2022) where each video is uniformly sampled into four temporal clips. For each clip, the shorter spatial side is resized to 224 pixels, and we then extract three $224 \times 224$ crops along the longer spatial axis. The final score is computed as the average across all views.

# H. Human Pose Tracking

The workflow is illustrated in Fig. 6. The output spike features from SpikeVideoFormer are passed through the regression head to predict parametric human poses over time, represented by the SMPL body model (Loper et al., 2015). In the regression head, we apply 2D average pooling, followed by three parallel linear layers to regress the SMPL shape parameters $\hat{\boldsymbol{\beta}} \in \mathbb{R}^{T \times 10}$, SMPL pose parameters $\hat{\boldsymbol{\theta}} \in \mathbb{R}^{T \times 72}$, and global translations $\hat{\mathbf{d}} \in \mathbb{R}^{T \times 3}$, respectively. Additionally, using predefined camera parameters, we can also compute the 3D and 2D joint positions.

**Training loss.** Following previous work (Zou et al., 2021), we use a combination of losses for the pose and shape pa-

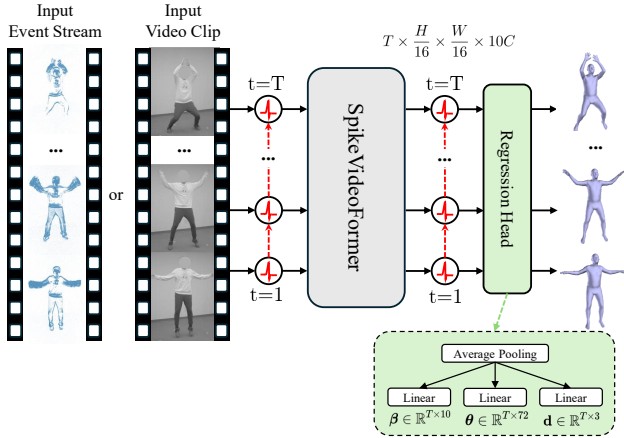

*Figure 6.* **Workflow of human pose tracking using our proposed SpikeVideoFormer.** The regression head consists of three parallel linear layers to regress the SMPL shape parameters.

rameters, global translations, and 3D and 2D joint positions to train our model. The total loss function is defined as:

$$\mathcal{L} = \lambda_{\text{pose}}\mathcal{L}_{\text{pose}} + \lambda_{\text{shape}}\mathcal{L}_{\text{shape}} + \lambda_{\text{trans}}\mathcal{L}_{\text{trans}} \quad (29)$$
$$+ \lambda_{\text{3D}}\mathcal{L}_{\text{3D}} + \lambda_{\text{2D}}\mathcal{L}_{\text{2D}},$$

where $\lambda_{\text{pose}}$, $\lambda_{\text{shape}}$, $\lambda_{\text{trans}}$, $\lambda_{\text{3D}}$, and $\lambda_{\text{2D}}$ are the respective loss weights. For the pose loss, we use the 6D representation of rotations. To measure the difference between the predicted and target poses, we compute the geodesic distance in $SO(3)$ as follows:

$$\mathcal{L}_{\text{pose}} = \sum_{t=1}^{T} \sum_{j=1}^{24} \arccos^2 \left( \frac{\text{Tr}\left(R^{\top}(\boldsymbol{\theta}_{[t]}^{j}) R(\hat{\boldsymbol{\theta}}_{[t]}^{j})\right) - 1}{2} \right),$$

where $R(\cdot)$ transforms the 6D rotational representation to the $3 \times 3$ rotation matrix, and $j$ is the joint index. For the other losses, we compute the Euclidean distances between the predicted and target values:

$$\mathcal{L}_{\text{shape}} = \sum_{t=1}^{T} \|\boldsymbol{\beta}_{[t]} - \hat{\boldsymbol{\beta}}_{[t]}\|^2, \quad \mathcal{L}_{\text{trans}} = \sum_{t=1}^{T} \|\mathbf{d}_{[t]} - \hat{\mathbf{d}}_{[t]}\|^2,$$

$$\mathcal{L}_{\text{3D}} = \sum_{t=1}^{T} \sum_{j=1}^{24} \|\mathbf{j}_{\text{3D}[t]}^{j} - \hat{\mathbf{j}}_{\text{3D}[t]}^{j}\|^2,$$

$$\mathcal{L}_{\text{2D}} = \sum_{t=1}^{T} \sum_{j=1}^{24} \|\mathbf{j}_{\text{2D}[t]}^{j} - \hat{\mathbf{j}}_{\text{2D}[t]}^{j}\|^2.$$

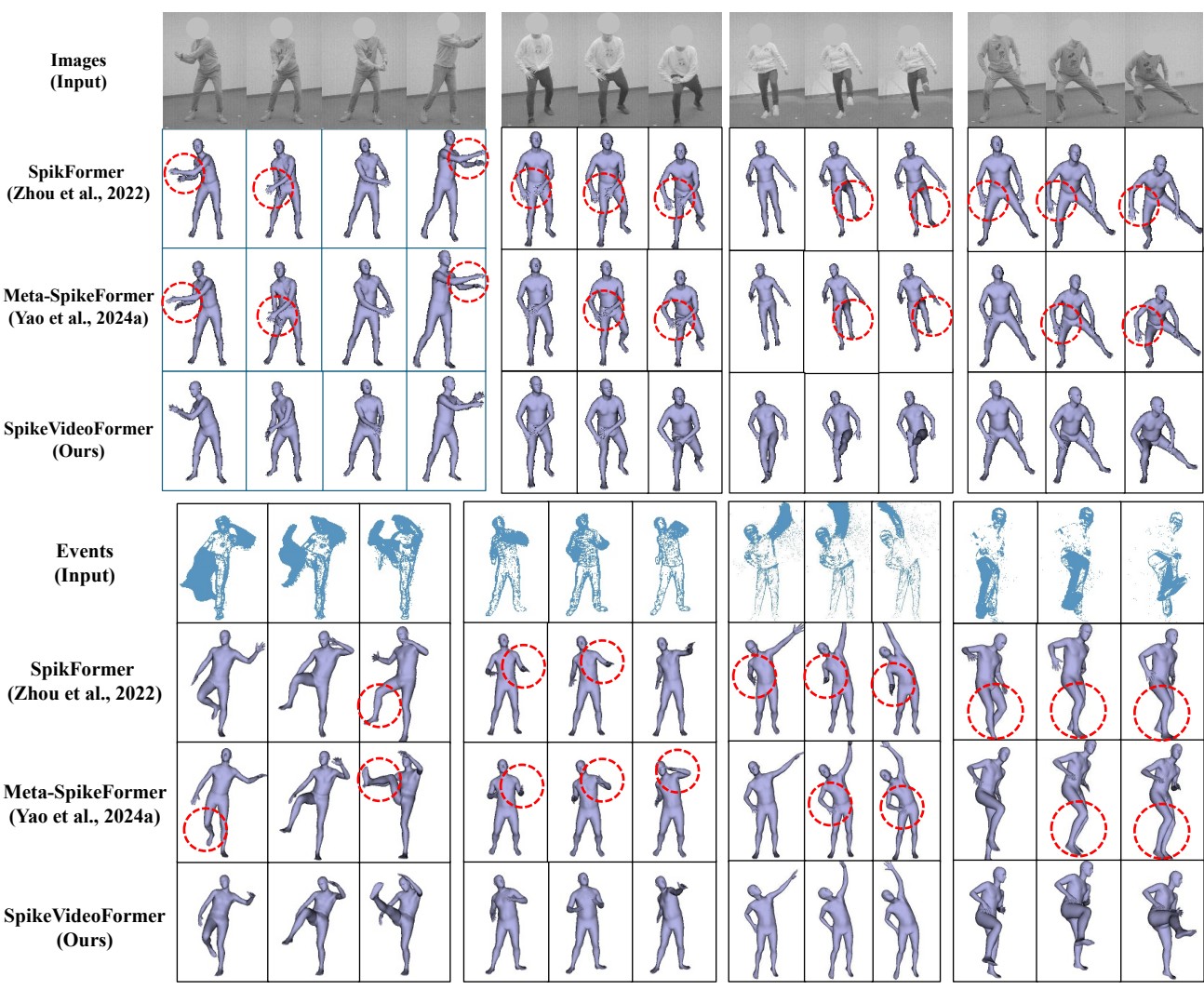

*Figure 7.* Qualitative Results of Our SpikeVideoFormer on Human Pose Tracking.

**Additional Implementation Details.** The input dimensions for both modalities are $T \times 256 \times 256 \times 3$. The event stream is evenly split into $3T$ segments over time, with each segment transformed into an event frame. In these frames, a pixel value of 0 indicates no events occurred at that position during the time period, while a pixel value of 1 signifies that events occurred at that position. These event frames are then resized to match the input shape required by the pre-trained SpikeVideoFormer.

For training, the $T = 8$ and $T = 32$ models are trained for 30 and 20 epochs, respectively, using a batch size of 64 and 8 across 8 NVIDIA A6000 GPUs. The learning rate is initially set to 6e-4 for the head and 6e-5 for the pre-trained backbone. The learning rate is adjusted by a CosineAnnealingLR schedule with a maximum of 35 and 25 epochs for the $T = 8$ and $T = 32$ models, respectively. To ensure robustness against both fast and slow motions, we apply two

types of augmentation: (1) randomly selecting video clips or event streams of varying durations—(1, 2, 3, 4) seconds for $T = 8$ and (4, 8, 16, 32) seconds for $T = 32$—with temporal frame stride of (1, 2, 3, 4) correspondingly; (2) applying a random spatial rotation to the input clip, ranging from -20 to 20 degrees. During testing, the durations are 2 and 16 seconds for $T = 8$ and $T = 32$ models, respectively.

## I. Video Semantic Segmentation

The workflow is illustrated in Fig. 9. Our SpikeVideoFormer encoder extracts four scales of spike features, each with the following dimensions: $T \times \frac{H}{2} \times \frac{W}{2} \times C$, $T \times \frac{H}{4} \times \frac{W}{4} \times 2C$, $T \times \frac{H}{8} \times \frac{W}{8} \times 4C$, and $T \times \frac{H}{16} \times \frac{W}{16} \times 2C$. These features are then separately forwarded into the Memory Read and Fuse modules. In this module, the feature from the last time step is used as the query, while the previous $T - 1$ time steps

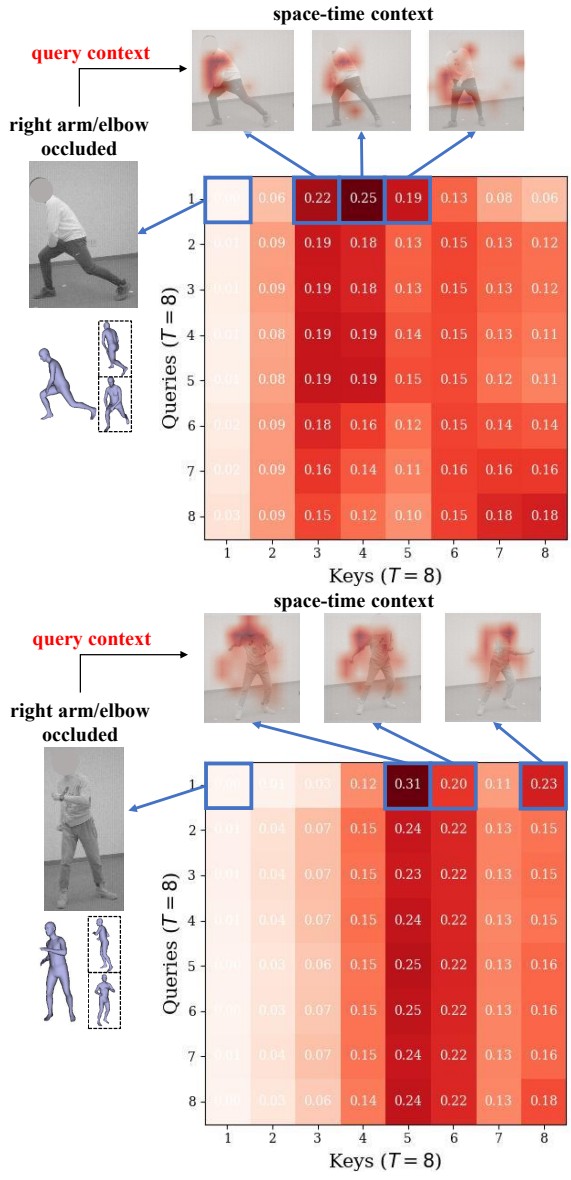

Figure 8. Visualization of Attention Maps in Human Pose Tracking

of features serve as the key and value for cross-attention via SDHA. The fused features of the last step across all four scales are subsequently used as pyramid features in the decoder SpikeFPN, which predicts pixel-level semantic categories.

For training, we use crop sizes of $480 \times 480$ for the VSPW dataset and $768 \times 768$ for CityScapes. During testing, crop sizes of $480 \times 863$ and $512 \times 1024$ are applied to these datasets, respectively. Data augmentation during training includes random resizing, flipping, cropping, and photometric distortion. The encoder and decoder are trained for 300 epochs on the CityScapes dataset and 60 epochs on the VSPW dataset, with a batch size of 16 across 8 NVIDIA

A6000 GPUs. The initial learning rate is set to 1e-3 and is adjusted using a CosineAnnealingLR schedule, with a maximum of 400 and 80 epochs for the two datasets, respectively. Finally, the Memory Read and Fusion module is trained for 100 epochs on CityScapes and 20 epochs on VSPW, with a learning rate of 1e-4.

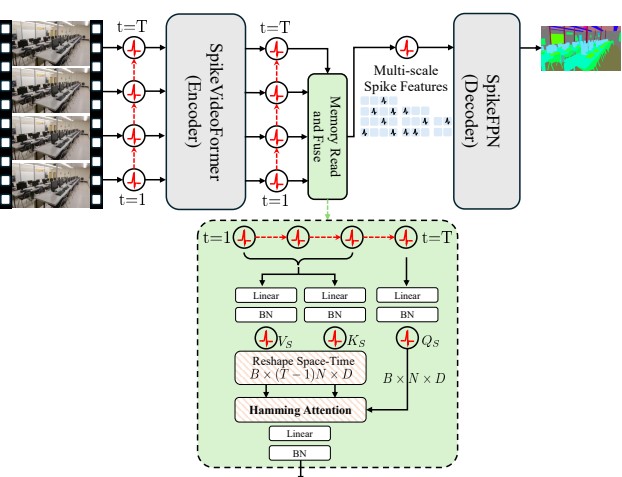

Figure 9. Pipeline of video semantic segmentation using SpikeVideoFormer as encoder, spike-driven Hamming attention as Memory Read and Fusion module, and SpikeFPN as decoder.

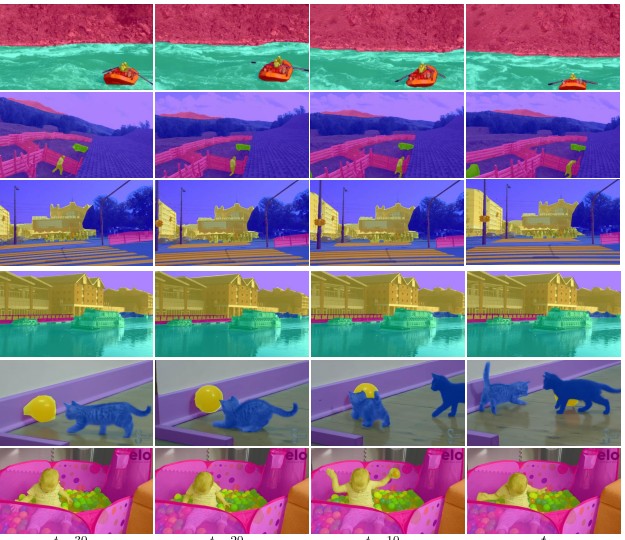

Figure 10. Qualitative Results of Our SpikeVideoFormer on VSS.

