# OpenReview forum: "SpikeVideoFormer: An Efficient Spike-Driven Video Transformer with Hamming Attention and $\mathcal{O}(T)$ Complexity"
_ICML.cc/2025/Conference — ICML 2025 poster_

### Official Review · Reviewer_ZrRF · 2025-03-12

**Overall Recommendation:** 3

**Summary:**

This manuscript introduces a video-based transformer model that implements spiking neural networks (SNNs) and Convolutional Neural Networks (CNN). The work highlights the efficiency of the proposed model in video-related tasks, particularly focusing on computational (parameters) and power efficiency.
A key contribution is Hamming Attention, a mechanism that solves the dot-product problem in the SNN within the attenrion mechanism, while maintaining linear computational scaling with respect to tokens (spatial/temporal dimensions) (O(TND²)).


The model is validated across three tasks:

-Human Pose Tracking

-Video Classification

-Video Semantic Segmentation

after rebuttal, I'll keep my recommendation

**Claims And Evidence:**

-Solving dot-product's problem in SNN, replacing this with the Hamming Attention

-Linear complexity with respect to tokens (spatial/temporal dimensions) (O(TND²))

-Validation of the model in three task, surpassing the State-Of-The-Art in the SNN category.

**Essential References Not Discussed:**

Object Detection's task is omitted

**Experimental Designs Or Analyses:**

No code or video is provided to veryffi the autentisity of the results

In attachments we can see some qualitative results comparing the task of human posse tracking.

The results of other models in the Video Semantic Segmentation task are not shown.

**Methods And Evaluation Criteria:**

-Dot product:

Because SNNs are sparse and do not have signals/elements at certain points (When Spike Query Contains No Elements), the dot product produces erratic Attension Maps.

-Lineal complexity:

It is demonstrated that the model scale linearly when longer sequences are processed

-The model is validated across three tasks:

Human Pose Tracking, Video Classification, Video Semantic Segmentation

**Other Comments Or Suggestions:**

No comments

**Other Strengths And Weaknesses:**

-Strengths

The manuscript is well-written and includes a wide variety of experiments supported by mathematical proofs.


-Weaknesses

While the manuscript focuses on developing spiking neural networks (SNNs), it omits addressing the task of Object Detection.
Also, recent advancements in transformers (ANN-based) for Semantic Segmentation are not adequately covered in the related work.

**Questions For Authors:**

No questions

**Relation To Broader Scientific Literature:**

While I am not deeply related, an improvement related to meta-spikeformer is made

Luo, X., Yao, M., Chou, Y., Xu, b., Andli, g. Integervalued Training and Spike-Driven Inference Spiking Neural Network for High-Performance and Energy-Efficien to Bject Detection.eccv, 2024.

However, Object Detection's task is omitted

**Theoretical Claims:**

-The derivative of Normalized Hamming Similarity is proved

-The proposed Hamming-based attention for SNN is proved

-The linear complexity with respect to the length of the tokens is experimentally proved

---

> ### Author Rebuttal · Authors · 2025-03-31
>
> Dear Reviewer ZrRF,
>
> We greatly appreciate your time and effort in reviewing our work. Below are our point-by-point responses to your comments.
>
> ---
>
> **Experimental Designs Or Analyses:**
>
> - Thanks for the constructive comment. For a qualitative comparison of video semantic segmentation, please refer to _Supp Figure 3_ on the [anonymous GitHub page](https://anonymous.4open.science/w/AnonymousSpikeVideoFormer-5E5A/). Additionally, video results are provided in _Supp Figures 5 and 6_. **The source code, results, and project website will be publicly released.**
>
> ---
>
> **Relation To Broader Scientific Literature** & **Essential References Not Discussed:**
>
> - **We have cited this paper [A] (Line 477) in our work** and also applied in the experiment of Video Semantic Segmentation (Line 386). Integer-valued spike representation proposed in the suggested paper represents spikes as a set of integers, e.g. {0,1,2,3} (Integer-LIF=4), rather than {0,1} only. During inference the integer spike can be separated as a sum of {0,1} spikes, e.g., 3= 1+1+1, 2=1+1+0. This separation maintains spike-driven efficiency of SNN methods, while improving the model's representation ability.
> - **We further compare our method on the object detection task** against this work and Meta-SpikeFormer, as shown in the table below. Following prior work, we use spiking-based YOLO as the detection head and evaluate on the COCO 2017 dataset. Our method surpasses Meta-SpikeFormer by 0.6 mAP but lags behind SpikeYOLO by 0.2 mAP. However, unlike SpikeYOLO, which is specifically designed for object detection, our method is more general and applicable to a wide range of downstream tasks.
>
> |Method|Param|Timestep|Integer-LIF|mAP@50|
> |:-|:-:|:-:|:-:|:-:|
> |SpikeYOLO [A]|13.2M|1|4|59.2|
> |Meta-SpikeFormer|16.8M|1|4|58.4|
> |SpikeVideoFormer (ours)|16.9M|1|4|59.0|
> |
>
> [A] Integer-valued Training and Spike-Driven Inference Spiking Neural Network for High-Performance and Energy-Efficient Object Detection. ECCV 2024.
>
> ---
>
> **Other Strengths And Weaknesses:**
>
> - Thanks for the valuable suggestion. Please refer to the response above under **Relation to Broader Scientific Literature** for the results related to the object detection task.
> - We present recent advancements in ANN-based Transformers for semantic segmentation as follows.
>
>   - Recent advancements in ANN-based Transformers have greatly enhanced semantic segmentation. SETR [1] pioneers a sequence-to-sequence approach, employing a pure Transformer encoder without convolutional layers. Segmenter [2] builds on a ViT backbone pre-trained on ImageNet and incorporates a mask transformer decoder to capture global context. SegFormer [3] further optimizes Transformer-based segmentation with a hierarchical encoder and a lightweight MLP decoder while eliminating positional encoding. Mask2Former [4] refines segmentation by restricting cross-attention to foreground regions using a masked attention operator. Recent CFFM [5] introduces coarse-to-fine feature assembling and cross-frame feature mining to capture both local and global temporal contexts. In contrast, we explore spiking video transformers to develop a faster, more energy-efficient approach for this task.
>
> [1] Rethinking semantic segmentation from a sequence-to-sequence perspective with transformers. CVPR 2021.
>
> [2] Segmenter: Transformer for semantic segmentation. ICCV 2021.
>
> [3] Pyramid vision transformer: A versatile backbone for dense prediction without convolutions. ICCV 2021.
>
> [4] Masked-attention mask transformer for universal image segmentation. CVPR 2022.
>
> [5] Learning local and global temporal contexts for video semantic segmentation. IEEE TPAMI 2024.
>
> ---
>
> We sincerely appreciate your feedback and will ensure that all results and discussions are thoroughly reflected in the final version.

---

### Official Review · Reviewer_4RFZ · 2025-03-12

**Overall Recommendation:** 4

**Summary:**

The authors present a novel model called the SpikeVideoFormer – a transformer network based on Spiking Neural Networks (SNN). They use Spike-Driven Hamming Attention (SDHA) instead of the usual dot product based self-attention. They claim their network to have a linear temporal complexity compared to the other model architectures that they explored. They show results on 3 tasks – video classification, video semantic segmentation and human pose tracking, achieving better performance compared to their ANN counterparts.

**Claims And Evidence:**

The authors made claims about the computational efficiency of their model with power measured in milli Joules instead of Watts, which is what would be expected when using large power systems like GPUs. They should do a comparison in terms of Watts, especially when comparing the ANNs.
Additionally, ANNs benefit from the matrix multiplications being sped up by GPUs, does the same hold true for SNNs? Authors should give insights/comparison along those lines as well.
The authors should give more specific insights into their training procedures w.r.t. hardware, compute costs, batch sizes,  etc.

**Essential References Not Discussed:**

Given the authors talk about the affinity of their network to the human brain, they should include relevant references on the cognitive science literature for fundamental tasks like visual object recognition, tracking, etc.

**Experimental Designs Or Analyses:**

The three tasks described in the paper for evaluating are set up well, besides my other comments related to reporting the metrics like watts and evaluating on tasks from cognitive psychology.

**Methods And Evaluation Criteria:**

I like the approach authors took in evaluating SpikeVideoFormer on a wide variety of tasks – segmentation, video understanding, human pose tracking. Given the SNNs ability to mimic the neuronal firing activity in the brain, they could also present a few tasks from cognitive psychology. For the primate visual system’s ability in tracking long range motion, they could evaluate their model on tasks like PathTracker for video or PathFinder for images.
An easier evaluation might be object recognition in noisy environments like Imagenet-C, and compare their methods against models of visual system like VOneNet or Extreme Image Transforms.

**Other Comments Or Suggestions:**

Some simplification of the writing to make the ideas more cogent would always help the reader.

**Other Strengths And Weaknesses:**

Authors show impressive results in terms of their performance compared ot the other state-of-the-art literature for ANNs and SNNs. One of the ways to strengthen the paper for the reader would be to show the internal workings of the network through some kind of spike-maps (akin to saliency maps) so that the reader can get insights into where the network focuses. This would give insights into some of the explainability metrics of the model.

**Questions For Authors:**

For long-term tracking in the human pose tracking, authors mention their results being close to GLoT in ANNs. Did the authors try to improve the results with something like a memory component or gating to improve the long-term tracking abilities of the model?

**Relation To Broader Scientific Literature:**

The authors present an important and novel architecture useful for a lot of downstream video related tasks. This work is relevant to the broader community of neuro-inspired computing with potential applications in long term tracking.

**Theoretical Claims:**

Proposition 3.1 is the theoretical aspect in the paper with its proof in the Appendix. On a broader look, it seems well written.

---

> ### Author Rebuttal · Authors · 2025-03-31
>
> Dear Reviewer 4RFZ,
>
> We sincerely appreciate your time and effort in reviewing our paper. Please find our point-by-point responses to your comments below.
>
> ---
>
> **Claims And Evidence:**
>
> - Thanks for the value suggestion. Normally, Power = Watts * Time. According to [B, C], when comparing ANNs and SNNs, we typically assume the hardware operates for the same duration. **Therefore, computational efficiency measured in power is equivalent to that measured in watts.** As suggested by other reviewers, we have also reported latency (inference time) as an additional evaluation metric for computational efficiency in our response to Reviewer 9Qge (W1 Inference Time).
> - ANNs rely on **floating-point multiplications**, where GPUs can accelerate in parallel but with high energy costs. In contrast, SNNs use **binary spikes to propagate, requiring only additions**—a special case (0/1 binary matrix) that GPUs can speed up as well. However, power consumption and processing time can be further reduced on specialized neuron-computing devices [A, B], where multiplication operator is replaced with much faster addition operators.
>
> [A] Neuromorphic computing at scale. Nature 2025.
>
> [B] Spike-based dynamic computing with asynchronous sensing-computing neuromorphic chip. Nature Communications 2024.
>
> [C] Firefly: A high-throughput hardware accelerator for spiking neural networks with efficient dsp and memory optimization. IEEE VLSI 2023.
>
> ---
>
> **Methods And Evaluation Criteria:**
>
> - **We evaluate our approach on the ImageNet-C dataset** to assess object recognition performance in noisy environments. Notably, Extreme Image Transforms [B] lacks ImageNet-C results, so we compare our method with VOneResNet50 [A] and ResNet50 [C].
>
> ||Noise|Gaussian|Shot|Impulse|Blur|Defocus|Glass|Motion|Zoom|
> |:-|:-:|:-:|:-:|:-:|:-|:-:|:-:|:-:|:-:|
> |ResNet50 (25.6M) [F]||29.6|27.9|24.5||38.8|26.1|37.9|34.5|
> |VOneResNet50 [D]||34.6|33.4|31.9||37.8|35.7|37.4|34.0|
> |Ours (15M)||33.1|34.8|38.0| |39.1|38.9|38.1|35.9|
> |
>
> ||Weather|Snow|Frost|Fog|Bright|Digital|Contrast|Elastic|Pixelate|JPEG|
> |:-|:-:|:-:|:-:|:-:|:-:|:-:|:-:|:-:|:-:|:-:|
> |ResNet50 (25.6M) [F]||30.1|36.7|43.6|66.8||38.8|44.8|47.0|55.1|
> |VOneResNet50 [D]||25.2|36.8|30.1|62.4||28.5|48.7|63.3|61.0|
> |Ours (15M)||28.3|37.0|42.1|66.2||34.5|49.3|61.4|61.8|
> |
>
> [D] Simulating a primary visual cortex at the front of CNNs improves robustness to image perturbations. NeurIPS 2020.
>
> [E] Extreme Image Transformations Facilitate Robust Latent Object Representations. arXiv 2023.
>
> [F] Deep residual learning for image recognition. CVPR 2016.
>
> ---
>
> **Supplementary Material** & **Other Strengths And Weaknesses:**
>
> - We have included ground truths for human pose tracking and the spike attention maps on the [anonymous github page](https://anonymous.4open.science/w/AnonymousSpikeVideoFormer-5E5A/).
>
> ---
>
> **Essential References Not Discussed:**
>
> - Visual cognitive neuroscience studies how factors like attention, motivation, emotion, and expectation shape visual perception and cognition [1]. While the first three enhance relevant stimuli processing, expectation suppresses predictable inputs [2-3]. Predictive processing suggests perception is inference-driven, refining sensory input through internal models shaped by context and experience [4]. Beyond vision, the visual cortex processes object names and activates in blind individuals, indicating broader cognitive roles in memory, imagery, and language [5-7]. Though its full scope remains debated, these insights inspire the brain-inspired spiking neural network (SNN) approach for complex visual tasks, enabling high-speed, low-energy neural processing.
>
> [1] The role of context in object recognition. Trends in Cognitive Sciences 2007.
>
> [2] How brains beware: neural mechanisms of emotional attention. Trends in Cognitive Sciences 2005.
>
> [3] Expectation (and attention) in visual cognition. Trends in Cognitive Sciences 2009.
>
> [4] An integrative, multiscale view on neural theories of consciousness. Neuron 2024.
>
> [5] Object domain and modality in the ventral visual pathway. Trends in Cognitive Sciences 2016.
>
> [6] The human imagination: the cognitive neuroscience of visual mental imagery. Nature Reviews Neuroscience 2019.
>
> [7] Reevaluating the sensory account of visual working memory storage. Trends in Cognitive Sciences 2017.
>
> ---
>
> **Questions For Authors:**
>
> - The performance gain stems from space-time joint attention. As shown in our ablation study (Table 6), using spatial-only attention results in pose tracking error (PA-MPJPE) increase of more than 36%. Moreover, GLoT is limited by the quadratic complexity of ANN-based attention, requiring image features to be represented as a high-level single vector. In contrast, our SNN's linear complexity allows for more detailed spatial and temporal feature fusion through spike-driven attention.
> ---
>
> We are grateful for your thoughtful comments and will carefully integrate all results and discussions into the final version.

---

> > ### Comment · Reviewer_4RFZ · 2025-04-04
> >
> > Thank you. I am pleased to see a more comprehensive literature review as part of the paper. For Extreme Image Transforms, the paper in Biological Cybernetics looks more cite worthy because of peer-reviews. I am also okay with the Power calculations and inference times as long as it makes it easier for the reader to understand the requirements. Please also include these are part of your revisions to the manuscript.

---

> > > ### Author Response · Authors · 2025-04-06
> > >
> > > Dear Reviewer 4RFZ, thank you for your encouraging and positive feedback. We are glad that your concerns have been addressed. We will refine our work according to your suggestions. In addition, our source code and project website will be made publicly available. Thank you again for your valuable time and effort in helping us improve our paper.

---

### Official Review · Reviewer_9Qge · 2025-03-13

**Overall Recommendation:** 3

**Summary:**

The authors propose SpikeVideoFormer, an efficient spike-based Transformer to process videos with linear temporal complexity. Technically, a spike-based Hamming attention mechanism is proposed from a theoretical perspective. Then, the authors further analyze several spike-based attention modules for video processing tasks. Finally, three typical video tasks (i.e., classification, human pose tracking, and semantic segmentation) are conducted to evaluate the effectiveness of the proposed method. The results show that the proposed SpikeVideoFormer outperforms SOTA methods in accuracy and energy efficiency.

**Claims And Evidence:**

Yes, the claims are very clear.

**Essential References Not Discussed:**

No

**Experimental Designs Or Analyses:**

The effectiveness of the proposed method has been extensively validated through numerous experiments; however, certain ablation studies warrant further exploration to provide deeper insights.

**Methods And Evaluation Criteria:**

Yes, the authors select three typical video tasks to verify the effectiveness of SpikeVideoFormer.

**Other Comments Or Suggestions:**

No

**Other Strengths And Weaknesses:**

Strengths:

1. Investigating low-power versions of Transformer architectures is a highly meaningful research direction.

2. The authors propose a spike-based Hamming attention mechanism and provide extensive theoretical proofs to support it.

3. The authors validate the effectiveness of SpikeVideoFormer on three downstream tasks and supplement the paper with detailed appendices for further clarification.


Weaknesses:

1. Inference Time Analysis. Although the authors provide a complexity analysis, the inference time of SpikeVideoFormer is not reported, including the inference times of several open-source baseline methods. It would be valuable to compare the inference times and discuss the feasibility of deploying SpikeVideoFormer-like architectures on edge devices. This analysis would significantly enhance the practical relevance of the work.

2. Real-World Event-based Vision Tasks. The selection of three typical video tasks is commendable. However, the authors do not explore real-world event camera tasks, such as long-term event stream processing, action recognition in real-world videos, or scene understanding. Including such tasks would further demonstrate the versatility and applicability of the proposed method in practical scenarios.

3. Energy Efficiency Analysis. While the authors reference previous methods to analyze the energy efficiency of SNN algorithms, the use of AC or MAC operations for power consumption calculations may not be entirely convincing for SNNs. Given the critical importance of this metric, the rationale behind this approach should be thoroughly justified. The authors are encouraged to provide their insights or at least discuss this limitation in detail, as it significantly impacts the credibility of the energy efficiency claims.

4. Ablation Studies and Parameter Analysis. The ablation experiments in the paper are relatively limited. Although the authors conduct three types of experiments, which may constrain the available space, it is essential to discuss the contributions of key parameters and modules. For instance, a more granular analysis of simulation time steps, hyperparameters would provide deeper insights into the proposed method's design and performance.

5. Some Clarity. The authors should clarify how SpikeVideoFormer differs from existing Video Transformers. Beyond replacing activation functions with binary spikes, what are the unique design elements of this novel Spike Transformer architecture? A detailed discussion on these aspects would help readers better understand the innovation and contributions of the proposed method.

**Questions For Authors:**

Please see the weaknesses and response each comment. Besides, two questions are listed below:

1. In Table 2, what is the difference between video and event stream as inputs for SpikeVideoFormer?

2. Why does the Spike-based Transformer structure emphasize video? Can it not process event streams? Doesn’t the title "SpikeVideoFormer" make the application seem too narrow?

**Relation To Broader Scientific Literature:**

Indeed, Transformers have become the primary architecture in deep learning. Investigating spike-based Transformers is particularly valuable from a power consumption standpoint. The authors introduce a new attention mechanism for video processing, advancing the field beyond prior approaches.

**Theoretical Claims:**

Yes，I have checked the correctness of all proofs.

---

> ### Author Rebuttal · Authors · 2025-03-31
>
> Dear Reviewer 9Qge,
>
> We are grateful for your insightful feedback. Below, we provide a detailed response to each of your points.
>
> ---
>
> **Other Strengths And Weaknesses:**
>
> ---
>
> **W1 Inference Time (per video clip $T\times 256\times 256\times 3$ as input)**
>
> - **We report the inference time in the table below**, tested on an A6000 GPU and AMD EPYC 7543 CPU for human pose tracking, averaged over 1,000 video clips with a batch size of 1. As $T$ increases from 8 to 32 (4x), GLoT (quadratic ANN attention) achieves the best performance but experiences a 9.8× increase in inference time, while VIBE (linear ANN-GRU) shows a 5.1x increase but performs the worst. Both SNN methods exhibit only 4.3× and 4.6× increases thanks to our proposed spiking space-time linear attention.
>
> |Method|Timestep|Power (mJ)|Inference Time (ms)|PA-MPJPE (mm)|
> |:-|:-|:-:|:-:|:-:|
> |VIBE (ANNs)| $T=8$ |392.1|**264**|46.3|
> ||$T =32$| 1511.2|**1335**|53.6|
> |GLoT (ANNs)| $T=8$ |487.5|**303**|39.9|
> ||$T =32$| 4046.1|**2972**|46.5|
> |Meta-SpikeFormer (SNNs)|$T=8$|95.7|**230**|45.7|
> ||$T =32$|387.2|**1001**|54.5|
> |SpikeVideoFormer (SNNs)|$T=8$|96.0|**235**|39.8|
> ||$T =32$|391.2|**1087**|47.5|
> |
>
> - **For edge device deployment**, SNNs, relying only on additions, significantly reduce power consumption and latency on neuron-computing devices [A, B]. ANNs, requiring floating-point multiplications, achieve high parallel acceleration on GPUs but at a high energy cost. On the AMD Xilinx ZCU104 [C], ANNs process at 691.2 GFLOPs/s, while SNNs reach 5529.6 GFLOPs/s—an 8x speedup. In 45nm technology [D], ANN multiplication consumes 4.6pJ, whereas SNN addition uses only 0.9pJ, a 5.1x energy reduction.
>
> [A] Neuromorphic computing at scale. Nature 2025.
>
> [B] Spike-based dynamic computing with asynchronous sensing-computing neuromorphic chip. Nature Communications 2024.
>
> [C] Firefly: A high-throughput hardware accelerator for spiking neural networks with efficient dsp and memory optimization. IEEE VLSI 2023.
>
> [D] 1.1 computing’s energy problem (and what we can do about it). IEEE ISSCC 2014.
>
> ---
>
> **W2 Event**
>
> - We have explored long-term event-based human pose tracking in Table 2.
> - We present results on event-based action recognition (HAR-DVS [E]) and event-based semantic segmentation for scene understanding (DDD17 [G]) in the table below.
>
> |Method|Param|Acc|
> |:-|:-:|:-:|
> |ACTION-Net [F] (ANNs)|27.9M|46.9|
> |Meta-SpikeFormer (SNNs)|15.0M|47.5|
> |SpikeVideoFormer (Ours)|15.0M|47.9|
> |
>
> |Method|Param|mIoU|
> |:-|:-:|:-:|
> |EV-SegNet [G] (ANNs)|13.7M|54.8|
> |SpikeVideoFormer (Ours)|17.8M|55.5|
> |
>
> [E] Hardvs: Revisiting human activity recognition with dynamic vision sensors. AAAI 2022.
>
> [F] Action-net: Multipath excitation for action recognition. CVPR 2021.
>
> [G] EV-SegNet: Semantic segmentation for event-based cameras. CVPR 2019.
>
> ---
>
> **W3 Energy**
>
> - Please refer to our response to W1 Inference Time. We have included latency (inference time) to evaluate efficiency in long-term tasks using our linear complexity method. Additionally, we have discussed the potential of deploying our SNN on specialized neurocomputing devices to further enhance its efficiency.
>
> ---
>
> **W4 Ablation**
>
> - We have conducted additional analysis of time steps and hyper-parameters on Human Pose Tracking (evaluated on PA-MPJPE). The results are shown in the tables below.
>
> |Timestep $T$|4|8|16|24|32|
> |:-|:-:|:-:|:-:|:-:|:-:|
> |PA-MPJPE|39.8|39.8|42.7|45.6|47.5|
> |
>
> |Channel size $C$|32|48|64|
> |:-|:-:|:-:|:-:|
> |Param|15.1M|31.3M|55.4M|
> |PA-MPJPE|44.4|41.7|39.8|
> |
>
> |Blocks|4-Transformer|1-CNN+3-Transformer|2-CNN+2-Transformer|3-CNN+1-Transformer|4-CNN|
> |:-|:-:|:-:|:-:|:-:|:-:|
> |Param|12.4M|13.8M|15.1M|16.5M|18.0M|
> |PA-MPJPE|40.7|40.1|39.8|45.6|54.9|
> |
>
> ---
>
> **W5 Clarity**
>
> - Our SpikeVideoFormer differs from existing Video Transformers in two aspects:
>   - Unlike **ANN-based Video Transformers**, which focus on **reducing quadratic complexity in space-time attention**, we highlight that spike-driven attention inherently achieves linear complexity (Table 1), making our approach more efficient and scalable.
>   - Unlike **existing SNN-based Transformers**, which focus on **single-image tasks only**, we introduce the first Spiking Video Transformer with an effective spike-driven attention mechanism (Proposition 3.1).
>
> ---
>
> **Questions For Authors:**
>
> - **Q1:** A video comprises a sequence of RGB images with 3×8 bits/pixel. In contrast, an event stream can be represented as a sequence of event frames with 1 bit/pixel (24x more sparse), indicating event occurrences within the time interval.
>
> - **Q2:** We emphasize videos due to their practicality in real-world applications, highlighting the need for efficient and effective processing methods. Following prior works, we also incorporate event-based tasks in our experiments to further demonstrate the applicability of SNNs.
> ---
>
> We are grateful for your suggestions and will ensure that the above results and discussions are reflected in the final revision.

---

### Official Review · Reviewer_oaZD · 2025-03-14

**Overall Recommendation:** 3

**Summary:**

The paper introduces SpikeVideoFormer, an efficient spike-driven video Transformer that leverages normalized Hamming similarity and joint space-time attention to achieve linear temporal complexity. It outperforms existing SNN-based models in video classification, human pose tracking, and video semantic segmentation while matching ANN-based methods in accuracy with significant efficiency gains.

**Claims And Evidence:**

The authors emphasize achieving linear temporal complexity with the proposed SpikeVideoFormer, but there does not appear to be a comparison in terms of latency to support this claim.

**Essential References Not Discussed:**

The necessary references are sufficiently cited in the paper.

**Experimental Designs Or Analyses:**

- The choice of SNN baselines, which are spiking transformers used for image processing, may raise questions from readers. If the authors propose a video-specific model, shouldn't they compare it with other spiking video transformers? If none exist, they should clearly state that theirs is the first.
- Among the three video-based vision tasks, space-time joint attention was applied to other Transformer-based SNNs only for human pose tracking (Table 2). Why is this considered a fair comparison? This approach does not seem consistent across other tasks.

**Methods And Evaluation Criteria:**

This work primarily proposes SDHA and space-time joint attention to enhance the performance of a spike-driven video transformer.
- Among them, SDHA appears to be a general method that is not limited to video transformers but can contribute to improving the performance of general SNN transformer architectures. In that case, could transformers for the image domain also benefit from SDHA? This is something I am curious about.
- On the other hand, space-time joint attention has already been used in existing ANNs, as mentioned in Section 3.4. Aside from replacing attention with SDHA, the novelty seems insufficient for it to be considered a main contribution. Could the authors further clarify the distinguishing aspects?

**Other Comments Or Suggestions:**

There are some formatting errors and typos in this paper.
- All table captions should be placed above the tables.
- Some notations for the equations in Section 3 appear to be missing.
- The figure on page 4 needs a caption.
- There are misplaced periods before and after a reference (p.2, line 99).

**Other Strengths And Weaknesses:**

Strengths
- The writing is clear and easy to understand.
- The use of Hamming similarity for attention scoring is interesting.
- The effectiveness of the method is demonstrated through various video-based vision tasks.

Weaknesses
- Please refer to the points mentioned in other sections.

**Questions For Authors:**

N/A

**Relation To Broader Scientific Literature:**

This work contributes to the broader scientific literature by expanding the role of spiking neural networks in the video domain.

**Theoretical Claims:**

I have checked the correctness of the proofs for the theoretical claims presented in the paper.

---

> ### Author Rebuttal · Authors · 2025-03-31
>
> Dear Reviewer oaZD,
>
> We appreciate your time and effort in reviewing our paper. Below, we provide a point-by-point response to your questions.
>
> ---
>
> **Claims And Evidence:**
>
> - **We report the latency in the table below**, based on tests conducted using the same hardware setup—a single A6000 GPU and an AMD EPYC 7543 CPU. The task is human pose tracking, comparing the performance of GLoT (ANNs) and our SpikeVideoFormer (SNNs). The input is an RGB video clip of shape $T\times 256\times 256\times 3$, with a batch size of 1. We conducted tests using 1,000 samples and averaged the time consumption as the latency. As the temporal length $T$ increases from 8 to 32 (4x), GLoT (quadratic attention) experiences a 9.8x latency increase, whereas SpikeVideoFormer (linear attention) shows only a 4.6x increase. GLoT’s increase is lower than the expected ~16× due to its use of a ResNet followed by a Transformer, where ResNet has linear complexity concerning temporal length.
>
> |Method|Timestep|Power (mJ)|Latency (ms)|PA-MPJPE (mm)|
> |:-|:-|:-:|:-:|:-:|
> |GLoT (ANNs)| $T=8$ |487.5|**303**|39.9|
> ||$T =32$| 4046.1|**2972 (9.8x)**|46.5|
> |SpikeVideoFormer (SNNs)|$T=8$|96.0|**235**|39.8|
> ||$T =32$|391.2|**1087 (4.6x)**|47.5|
> |
>
> ---
>
> **Methods And Evaluation Criteria:**
>
> - **Q1**: **Yes, our proposed SDHA can also benefit spike-driven transformers in the image domain.** As demonstrated in the table below (also in Appendix F, Table 8), applying SDHA leads to an accuracy improvement of 0.4% (15.1M model) and 0.2% (55.4M model) on ImageNet. In this experiment, our method follows the same model architecture as Meta-SpikeFormer, except for the inclusion of SDHA.
>
> |Method|Attention|Param|Power(mJ)|Top-1 Accuracy|
> |:-|:-:|:-:|:-:|:-:|
> |Meta-SpikeFormer|SDSA|15.1|16.7|73.2|
> |Ours|**SDHA**|15.1|16.8|73.6 (**+0.4**)|
> |Meta-SpikeFormer|SDSA|55.4|52.4|79.7|
> |Ours|**SDHA**|55.4|52.6|79.9 (**+0.2**)|
> |
>
> - **Q2**: We respectfully believe that our proposed method offers valuable contributions and introduces new insights compared to previous ANN-based works, specifically:
>   - Space-time joint attention in ANNs suffers from quadratic computational complexity, and most related methods propose different space-time attention designs to reduce this complexity (as shown in Table 1). However, to our knowledge, **we are the first attempt to show that space-time spiking attention designs share the same linear complexity.** The experiments demonstrate that our spike-driven solution achieves performance comparable to recent ANN-based methods while offering significant efficiency gains, with improvements of x16, x10, and x5 on three video-based tasks.
>   - **Our primary contribution lies in the design of the first Spiking Video Transformer** with effective spike-driven attention design (Proposition 3.1). This approach achieves SOTA performance compared to existing SNN approaches, with over 15% improvement on human pose tracking and video semantic segmentation.
>
> Thanks for the helpful questions. We will clarify the uniqueness and novelty in the final version.
>
> ---
>
> **Experimental Designs Or Analyses:**
>
> - **Q1**: We are sorry that the statement of our method is unclear than intended. **To our knowledge, we are the first to explore spiking video Transformer.** Thanks for the constructive comment. We will clarify the novelty in the final version.
>
> - **Q2**: **Other Transformer-based SNNs are image-based approaches** that only use spatial attention. For a fair comparison on video-based tasks, we adapted these approaches to incorporate space-time joint attention. This setting is applied consistently across all three tasks in our work. We will clarify this point in the final version to avoid confusion.
>
> ---
>
> **Supplementary Material:**
>
> - Thanks for pointing out this issue. We will include the supplementary materials as an appendix at the end of the main manuscript in the revised submission.
>
> ---
>
> **Other Comments Or Suggestions:**
>
> - We will carefully address the formatting errors in the table captions, correct the notations in the equations, and fix any typos in the final version.
>
> ---
>
> We value your feedback and will ensure that the above results and discussions are thoroughly addressed in the final revision.

---

### Decision · Program_Chairs · 2025-05-01

**Decision:**

Accept (poster)

**Comment:**

This paper received consistently positive reviews. The authors did an excellent job addressing the initial concerns raised by the reviewers through a clear and thorough rebuttal. The AC concurs with all reviewers and appreciate the thoughtful effort put into the response.

In addition, the AC believes this paper makes a solid contribution, presenting a novel method supported by impressive results. Therefore, the paper is recommended for acceptance. Please ensure that the clarifications provided in the rebuttal are incorporated into the camera-ready version to further strengthen the final submission.